

# Estimating the disequilibrium in denudation rates due to divide migration at the scale of river basins.

Timothée Sassolas-Serrayet[1], Rodolphe Cattin[1], Matthieu Ferry[1], Vincent Godard[2], Martine Simoes[3]

[1]Géosciences Montpellier, Université de Montpellier and CNRS UMR 5243, Montpellier 34095, France.
[2]Aix Marseille Univ, CNRS, IRD, INRA, Coll France, CEREGE, Aix-en-Provence, France
[3]Université de Paris, Institut de physique du globe de Paris, CNRS, F-75005 Paris, France

*Correspondence to*: Timothée Sassolas-Serrayet (timothee.sassolas-serrayet@umontpellier.fr)

**Abstract.** Basin-averaged denudation rates may locally exhibit a wide dispersion, even in areas where the topographic steady state is supposedly achieved regionally. This dispersion is often attributed to the accuracy of the data or to some

degree of natural variability of the signal, but it can also be attributed to stochastic processes such as landsliding. Another physical explanation to this dispersion is that of local and transient disequilibrium between tectonic forcing and erosion at the scale of catchments. Recent works have shown that divide migration can potentially induce such perturbations and they propose reliable metrics to assess divide mobility based on cross-divide contrasts in headwater topographic features. Here, we use a set of landscape evolution models assuming spatially uniform uplift, rock strength and rainfall to assess the effect

of divide mobility on basin-wide denudation rates. We propose the use of basin-averaged aggressivity metrics based on cross-divide contrasts in channel-head $\chi$, in local slope and in elevation. From our simulations, we show that the metric based on differences in channel-head elevations across divides is the least reliable to diagnose local disequilibrium. For the other metrics, our results suggest a nonlinear relationship with the ratio of basin denudation to uplift, which can reach up to a factor of two, regardless of the imposed uplift rate, diffusivity coefficient and critical hillslope gradient. A comparison with

field observations in the Great Smoky Mountains (southern Appalachians, USA) underlines the difficulty of using the metric based on $\chi$, which depends on the - poorly constrained - elevation of the outlet of the investigated catchment. Regardless of the considered metrics, we show that observed dispersion is controlled by catchment size: a smaller basin may be more sensitive to divide migration and hence to disequilibrium. Our results thus highlight the relevance of divide stability analysis from digital elevation models as a fundamental preliminary step for basin-wide denudation rate studies based on cosmogenic

radionuclide concentrations.

## 1 Introduction

Topographic steady state, in which average topography is constant over time, is one of the key concepts of modern geomorphology (e.g. Gilbert, 1877; Hack 1960; Montgomery, 2001). Though simple, this paradigm provides a useful framework to study landscape evolution related to tectonic and/or climatic forcing (e.g. Willett et al., 2001; Reinhart and

Ellis, 2015), to spatial variations in rock strength (Perne et al., 2017) or to the geometry of active crustal structures (Lave and



Avouac, 2001; Stolar et al., 2007; Scherler et al., 2014; Le Roux-Mallouf et al., 2015). To define topographic steady state, the temporal and spatial scales of the processes involved are essential parameters. Compared to large scale geodynamic processes operating over 1-100 Myr timescales, river incision and sediment transport are rapid processes driving landscapes to stable forms over this long timescale, whereas rapid climatic fluctuations during the Quaternary may prevent modern

landscapes from reaching steady-state conditions in modern landscapes (Whipple, 2001).

The timescale of river divide migration has received increasing attention in the recent years. Although rivers exhibit a rapid adjustment to tectonic or climatic changes to maintain their profiles, Whipple et al. (2017) show that divides continue to migrate over time periods of $10^6$-$10^7$ years, suggesting that long-term transience might be pervasive in the planar structure of landscapes (Hasbargen and Paola, 2000; Hasbargen and Paola, 2003; Pelletier, 2004). In addition to the influence of spatial

variability of rock uplift rate, rock strength or rainfall (e.g. Reiner et al., 2003; Godard et al., 2006; Miller et al., 2013), this long timescale could also explain the persistence of spatial variations in denudation rates observed in tectonically inactive orogens in spite of a supposedly and theoretically topographic steady state is achieved (Willett et al., 2014).

As an example of this, in the Great Smoky Mountains in the southern Appalachians, uplift and erosion rates integrated over varying time periods from 10s kyr to 100 Myr give a similar average magnitude of 0.025-0.030 mm.yr$^{-1}$. These results

suggest a regional quasi-topographic steady state over the last ~180 Myr, maintained by the isostatic response of the thickened crust since the end of the Appalachian orogeny (Matmon et al., 2003 a,b). Beyond this average value, individual basin-wide denudation rates exhibit a strong dispersion (up to a factor of two, Fig. 1), which is not related to spatial variation in rainfall or in erodibility of the substrate (Matmon et al., 2003b). In a recent study, Willett et al. (2014) assess divide mobility from the contrast in the channel head topographic metric $\chi$, taken here as a proxy for steady-state river profile

elevation (Perron and Royden, 2012; Royden and Perron, 2013), and propose an explanation in which a significant part of the observed dispersion in denudation rates could be due to drainage divide migration associated with contrasting erosion rates across divides.

Divide migration is often assessed through the metric $\chi$ (Willett et al, 2014). More recently, Forte and Whipple (2018) introduced other metrics, referred as "Gilbert metrics" (Gilbert, 1877), based on the cross-divide contrast in channel head

local slope and elevation in order to characterize divide migrations. This last study indeed focused on cross-divide contrasts in headwater basin shape. Here, we model divide migrations and propose new metrics of divide stability at the scale of the entire watershed, which are an expansion of the aggressivity metric initially suggested by Willett et al. (2014). We use these metrics to assess the effect of persistent divide mobility on basin-averaged erosion rates at a timescale of $10^4$ yr. We use numerical landscape evolution models, taking into account both hillslope diffusion and fluvial incision. For the sake of

simplicity and to avoid the influence of other factors such as topography, lithology, climate or vegetation, we restrict our analysis to synthetic orogens with spatially uniform uplift, rock strength and rainfall. After a brief presentation of the used landscape evolution model (LEM), we describe the methods developed to assess basin-wide denudation rates and aggressivity metrics, such as average cross-divide contrasts in headwater $\chi$, slope and elevation. Next, we investigate transient time and location of morphologic adjustments to divide migrations. We explore the relevance and complementarity





of tested relative stability metrics between neighbouring basins. We then investigate the impact of uplift rate and hillslope process on the dynamics of divide migration and associated denudation rates. Finally, we apply our approach to the basin-wide denudation rates dataset of Matmon (2003a,b) in the case of the Great Smoky Mountains and propose new criteria to guide future sampling strategies to assess basin-wide denudation rates from river sands.

## 2 Methods

### 2.1 Landscape Evolution Model (LEM)

We use TTLEM (TopoToolbox Landscape Evolution Model) (Campforts et al., 2017), a landscape evolution model based on the Matlab function library TopoToolbox 2 (Schwanghart and Scherler, 2014). This LEM uses a finite volume method (Campforts and Govers, 2015) to solve the following equation of mass conservation for rock/regolith subject to uplift and denudation:


$$\frac{\partial z}{\partial t} = \left(\frac{\partial z}{\partial t}\right)_{td} + \begin{cases} U + \left(\frac{\partial z}{\partial t}\right)_{fluv} & \text{for } A < A_c \\ \frac{\rho_r}{\rho_s}U + \left(\frac{\partial z}{\partial t}\right)_{hill} & \text{for } A < A_c \end{cases},$$   (1)

where $\partial z/\partial t$ is the variation of elevation with time, $(\partial z/\partial t)_{td}$ is the change of elevation due to tectonic horizontal advection, $U$ is the rock uplift rate, $\rho_r/\rho_s$ is the density ratio between the bedrock and the regolith, $A$ is the upstream area

and $A_c$ is a critical drainage area which corresponds to the transition between hillslope and fluvial processes.

Hillslope denudation is given by a non-linear formulation (Roering et al., 1999):

$$\left(\frac{\partial z}{\partial t}\right)_{hill} = -\nabla q_s \qquad \text{with} \qquad q_s = -\frac{D \nabla z}{1-\left(\frac{|\nabla z|}{S_c}\right)^2},$$   (2)

where $q_s$ is the flux of soil-regolith material. This flux rate increases to infinity when slope tends to a critical value $S_c$. The diffusivity $D$ gives the rate of soil-regolith material creep. Its magnitude ranges from $10^{-3}$ to $10^{-1}$ m².yr$^{-1}$ in natural settings and varies with soil thickness, lithology and vegetation (Roering et al., 1999; Jungers et al., 2009; West et al., 2013; Richardson et al., 2019). Hillslope diffusion is implemented in TTLEM using an implicit scheme, which is unconditionally stable at large time steps (Perron, 2011).

Fluvial incision is calculated with a stream power law:

$$\left(\frac{\partial z}{\partial t}\right)_{fluv} = -KA^m \left(\frac{\partial z}{\partial x_\Gamma}\right)^n,$$   (3)



$K$ is the erodibility coefficient reflecting climate, hydraulic roughness, sediment load and lithology. Its value ranges between

$10^{-16}$ and $10^{0}$ m$^{(1-2m)}$.yr$^{-1}$ (Kirby and Whipple, 2001; Harel et al., 2016). $A$ is the upstream area. $x_{\Gamma}$ is the along stream distance from the outlet of the river. $m$ and $n$ are two parameters which are usually reported as a $m/n$ ratio ranging between 0.35 and 0.8. River incision law is implemented in TTLEM using an explicit scheme based on a higher-order flux-limiting finite volume method that is total variation diminishing (TVD-FVM) [see Campforts and Govers (2015) and Campforts et al. (2017) for further details]. Its main advantage is to eliminate numerical diffusion, which is present in most other schemes

solving differential equations of river incision. This last point has a significant impact on the accuracy of basin-wide simulated denudation rates, making TTLEM a well-suited LEM for the purpose of this study.

### 2.2 Modeling approach and assumptions

#### 2.2.1 Geometry and meshing

Since the computation is performed using a discretized land surface, smaller mesh sizes will lead to detailed topography but

will lengthen the computation time and memory requirements. Hereinafter, we consider a reference square landscape model of 50 km side with a grid resolution of 90 m, which is a good compromise between computation time (3-5 hours on a PC workstation) and the total amount of basins that can be studied (>1000). Our results are not affected when the mesh resolution is increased to 30 m nor are they when the model size is multiplied by four (100x100 km) (See Fig. S1).

#### 2.2.2 Boundary conditions

In order to isolate the effect of divide migrations on the variability of basin-wide denudation rates, we explore simple models with constant and spatially uniform uplift and precipitation rates and we assume no horizontal advection $(\partial z/\partial t)_{td} = 0$. We use a Dirichlet boundary condition: simulation edges are not affected by uplift on a one pixel band to represent a stable base level for rivers. The model presents no initial topography, except for gaussian noise ranging between 0 and 50 m so as to initiate a random fluvial network.

#### 2.2.3 Set of parameters

Firstly, we consider a reference model with parameters commonly used for moderately active orogens: an uplift rate $U$ of 1 mm.yr$^{-1}$, a diffusivity $D$ of $10^{-2}$ m$^{2}$.yr$^{-1}$ (Roering et al., 1999), a threshold slope $S_c$ of 30° (Burbank et al., 1996; Montgomery et Brandon, 2002; Binnie et al., 2007), a $m/n$ ratio of 0.5 with $m = 0.5$ and $n = 1$, an erodibility coefficient $K$ of $5\times10^{-6}$ m$^{1-2m}$.yr$^{-1}$, a $\rho_r/\rho_s$ ratio of 1.3 and a critical drainage area $A_c$ of 0.2 km² (Montgomery et al., 1993).

Secondly, all other parameters held constant, we investigate the specific impact of uplift rate and hillslope processes in other models by varying $U$, $D$ and $S_c$ between 0.5 and 2 mm.yr$^{-1}$, $10^{-3}$ and $10^{-1}$ m$^{2}$.yr$^{-1}$ and 20° and 40°, respectively.



In order to better constrain the variability of our results under similar conditions, we run for each model five simulations using the same parameters, but with different initial random topographies.

### 2.2.4. Timescale

The total duration of simulations is 10 Myr, which is nearly one order of magnitude longer than the theoretical time to reach general topographic equilibrium for our set of parameters (Willett, 2001). The implicit scheme used to simulate non-linear hillslope processes provides stable solutions regardless of the time step. In contrast, the explicit scheme used to model fluvial incision requires a time step that satisfies the Courant-Friedrich-Lewy criterion. Hereinafter, we choose a time step of 500 yr for hillslope diffusion. Incision computation is nested in this time step and uses another time step that is automatically

determined to assure model stability (Campforts et al., 2017).

### 2.3 Basin-wide denudation rates and aggressivity metrics

### 2.3.1 Basin-wide denudation rates

We derive basins from the synthetic DEMs (Digital Elevation Models) using an accumulation map computed with a single flow direction algorithm implemented in TopoToolbox (Schwanghart and Scherler, 2014). Next, we calculate for each basin

the variation in elevation over a time interval of 10 kyr, which averages the results over 20 time steps. The drainage network can migrate during the simulation, so we only survey the basins that keep the same outlet location during this time interval. Furthermore due to divide mobility, the geometry of watersheds can also change. Hence, we measure the average difference in elevation inside the basin perimeter after 10 kyr. Here we only assess the surface uplift $U_s$ (England and Molnar, 1990). To obtain the real denudation rates for each basin, we sum $U_s$ with the uplift rate $U$ assumed in our simulation and divide the

result by the time interval, which is an approximation. However, calculating incremental denudation rates over each time step is prohibitive in terms of computation time for the number of extracted basins. By considering the relatively small period over which we integrate denudation (10 kyr), we then assume that these approximations have a negligible impact on the results at first order. Calculated that way, the denudation rate is sensitive to divide migration but also to transient features like knickpoints that migrate along the river network. We use the knickpointfinder algorithm implemented in TopoToolbox

(Schwanghart and Scherler, 2014) to identify the affected basins.

### 2.3.2 From cross-divide metrics to basin averaged aggressivity metrics

Most recent studies have focused on the relationship between drainage divide mobility and headwater across-divide contrast in either $\chi$, slope, elevation or local relief values (e.g. Whipple et al., 2017; Forte and Whipple, 2018). Here, in line with Willett et al. (2014, see Supp. Mat. therein) we focus on the specific influence of divide migration on denudation rates at the

scale of the entire stream basin. Our approach aims to integrate cross-divide contrasts in drainage network properties along



the entire basin perimeter. We then obtain basin-averaged aggressivity metrics that determine if a watershed is either growing or shrinking (Willett et al. 2014).

First we calculate $\chi$, local slope and elevation map for each channel pixel. $\chi$ is an integral function of position along the channel network (Perron and Royden, 2012) described by the equation :


$$\chi = \int_{x_b}^{x} \left( \frac{A_0}{A(x)} \right)^{\frac{m}{n}} dx \; , \tag{4}$$

where $A(x)$ is the upstream drainage area at the location $x$, $A_0$ is an arbitrary scaling area set to 1 km². The $m$ over $n$ ratio refers here to the reference concavity of an equilibrated river profile. Its value is set to 0.5 in accordance with the model

parameters. For each independent drainage network, we integrate $\chi$ from the outlet $x_b$, located at the model boundary (< 1 m high), to the channel heads. We then calculate the difference in channel-head $\chi$, local slope and elevation across each first order basin divide segment. The aggressivity metric is finally obtained by averaging these first order across-divide differences along the perimeter of each extracted basin (Fig. S2). This way, the sign of the aggressivity metric in a basin corresponds to the difference of the averaged value of considered metric (Channel head $\chi$, slope or elevation) in this basin

with respect to his neighbours. This method has the advantage to ponderate the weight of individual divide segments by the number of pixels they contain which provide an accurate assessment of the basin aggressivity.

## 3 Results

### 3.1 Evolution of reference model

A detailed analysis of the DEM suggests that during the initial phase, the flat initial surface (Fig.2a) is progressively uplifted

to form a plateau. At the same time the edges of this plateau are gradually regressively eroded by drainage networks that spread from the base level toward the center of the model (Figs. 2b and c). This transient landscape is completely dissected after 2 Myr. From this time and until the end of the simulation, landscape changes are mainly due to competition between watersheds, resulting in continuous divide migrations with decreasing intensity as the model is moving toward a total topographic equilibrium (Fig. 2d to f; Supplementary Video n°1).

We measure the average elevation, the maximum elevation and the average denudation rate over the entire model for each time step (Fig. 3a) and identify two distinct stages during the evolution of our reference simulation. During the first million years, due to long wavelength topographic building, the calculated landscapes are far from steady state. This leads to a major increase of the mean elevation from 0 m to ca. 700 m. In a second stage, this trend reverses and the mean elevation decreases asymptotically toward ca. 600 m until the end of the simulation.





The evolution of the maximum elevation follows the same pattern but can be affected by temporal changes in the location and altitude of higher peaks. The maximum elevation increases between 0 and ca. 2200 m over the first 2 Myr (Figs. 3a) then decreases progressively to reach ca. 1600 m at the end of the simulation.

We compute the average denudation rate from the tectonic uplift and from average elevation change over the entire model between two time steps:


$$(\Delta z/\Delta t)_{av} = U - E_{av} \, , \tag{5}$$

where $(\Delta z/\Delta t)_{av}$ is the average surface uplift over the entire model on a time-step $\Delta t$, $U$ is the imposed uniform uplift rate (1 mm.yr$^{-1}$) and $E_{av}$ is the average real denudation rate. During the first 1 Myr, the mean denudation rate increases but
remains lower than the uplift rate, leading to the increase in average elevation over this time period. Next in the following 0.5 Myr $E_{av}$ exceeds the uplift rate to reach up to 1.03 mm.yr$^{-1}$. It then gently decreases to 1 mm.yr$^{-1}$ during the rest of the simulation. This shows that topography tends to - but never reaches - a strict steady state over the simulation time. Abrupt changes in $E_{av}$ after ca. 2 Myr, 2.5 Myr and 3.2 Myr can also be highlighted (Fig. 3b). These brief variations are related to major local captures in the drainage network, which can be observed during the model evolution (Fig. 2d and Supplementary
Video n°1).

On the basis of these results, we will consider that quasi-topographic steady-state is reached between 1.5 and 2 Ma, when the plateau relict topography is totally eroded and $E_{av}$ begins to decrease (Figs. 2 and 3). This time is consistent with the time required to reach topographic steady state proposed from models with constant uplift rate and no horizontal advection (Willett et al., 2001).

**3.2 Basin-wide denudation rates variability**

We calculate basin-wide denudation rates $E$ upstream of each stable drainage network confluence after 2.5 Myr, 5 Myr and 10 Myr of simulation (Figs. 4a, b and c). As explained in the method section, we compiled the results obtained for five runs using the same parameters as the reference model, in order to increase the number of sampled basins. Regardless of the duration, we observe a significant variability in the calculated denudation rates depending on basin size. This variability is
maximum for small basins (ca. 1 km²) and decreases with increasing basin area. In our approach, small basins are nested in larger ones. Hence, these results can be related to the averaging of denudation rates along the drainage network, in agreement with the measurements of Matmon et al. (2003b). This variability also decreases with time (Figs. 4a, b and c). For basins with an excess of denudation relative to the uplift rate $U$, the $E/U$ ratio can reach up to 3 after 2.5 Myr but only 2 after 5 Myr and 1.5 after 10 Myr. For basins with a deficit of denudation, this ratio can be lower than 0.25 after 2.5 Myr, but increases to
0.4 after 5 Myr and 0.5 after 10 Myr. To assess more accurately the temporal evolution of this variability, we calculate $E$ every 0.5 Myr for three distinct categories of basin sizes: 1-2 km$^2$, 10-20 km$^2$ and 100-200 km$^2$. We then estimate the mean absolute deviation (MAD) from the uplift rate by considering separately basins with a denudation in excess or in deficit (Fig.





4d). Until 1.5 Ma, basins are located on the plateau where denudation rate is null. This leads to a low MAD for basins with a denudation deficit and to the absence of basins with a denudation excess. After 1.5 Ma, basins in deficit exhibit an asymptotic increase in MAD from nearly -0.2 to -0.05 mm.yr$^{-1}$, regardless of the area class considered. For basins in excess, the MAD value decreases through time, depending on drainage area : from ca. 0.28 mm.yr$^{-1}$ to ca. 0.05 mm.yr$^{-1}$ for basins with an area of 1-2 km² and 10-20 km²; from ca. 0.14 mm.yr$^{-1}$ to ca. 0.05 mm.yr$^{-1}$ for basins with an area of 100-200km². These results reflect a significant spatial variability of the difference between basin-wide denudation rates and uplift rate in our reference models. We also see a coherent evolution of this difference over the simulation time, consistent with the model progression toward a total topographic equilibrium.

The spatial variability of the denudation rates is neither homogeneous nor randomly distributed (Fig. 5a). The location of drainage basins with denudation rates far from the equilibrium value of 1 mm.yr$^{-1}$ coincides with migrating drainage divides (Fig. 2e) and with cross-divide contrasts in headwater $\chi$, slope and elevation (Figs. 5b, c and d). Following Willett et al. (2014) and Forte and Whipple (2018), the divide migrations predicted by these contrasts are consistent with the direction of divide mobility obtained from our model. One may note that the higher the contrast in these parameters across the divide, the higher the deviation of the denudation rate from the uplift rate, and therefore from topographic equilibrium. These results based on simulations assuming uniform and constant properties as well as constant boundary conditions confirm that the dispersion observed in denudation rates is primarily controlled by divide migration. Basins that expand (shrink) show higher (lower) denudation rates compared to uplift rate, and are hereafter referred to as aggressors (victims), following the terminology adopted by Willett et al. (2014).

### 3.3 Deviation of denudation rates from the uplift rate, and basin aggressivity

Willett et al. (2014) showed that the basin-averaged cross-divide contrast in $\chi$, could be used to deduce an aggressivity metric for basins. We extend this basin-scale approach to the Gilbert's metrics recently proposed by Forte and Whipple (2018) including cross-divide contrast in headwater slope and elevation. Hereinafter, we refer to these aggressivity metrics based on cross-divide contrast in headwater $\chi$, headwater slope (gradient) and headwater elevation (height) as $\Delta\chi$, $\Delta G$ and $\Delta H$, respectively.

We here assess the relationship between the $E/U$ ratio and these aggressivity metrics. First, to exclude variability related to both basin area and time, we focus on a single class of basins with a size of 10-20 km² gathered from five computed reference models after a simulation duration of 2.5 Myr. In agreement with cross-divide metrics tested by Forte and Whipple (2018), aggressor (victim) basins have negative (positive) $\Delta\chi$ and $\Delta H$ values and conversely positive (negative) $\Delta G$ value. Theoretically, aggressor (victim) basins have higher (lower) denudation rates than the underlying uplift rate. Therefore graphs in figure 6 must be divided into four quadrants, with aggressors situated in the lower-left one, and victims in the higher-right one. This result is verified for ca. 91, 88 and 82 % of basins for aggressivity metric based on headwater $\chi$, slope and elevation values, respectively (Figs. 6a, b and c). Several basins depart significantly from the expected quadrants for $\Delta G$ and $\Delta H$: these exhibit significant knickpoints in their drainage network that increase measured denudation rates. For this





limited dataset, the evolution between $E/U$ and both $\Delta\chi$ and $\Delta G$ can be defined by a linear relationship (Figs. 6a and b). Part of the dispersion observed around this first-order trend may be explained by approximations in the calculation of denudation rates and aggressivity metrics. Compared to other metrics, $\Delta H$ seems to be less sensitive to drainage migration and shows a more scattered distribution (Fig. 6c).

In natural settings, the stage of evolution of landscapes cannot be easily defined and the total amount of basins with a specific size may be limited. The large dataset provided by our model can provide further insights by gathering the results obtained every 0.5 Myr for seven classes of basin areas expanding geometrically with a multiplying factor of 2 from 1-2 to 64-128 km² (Figs. 6d, e and f). Denudation rates can be affected by knickpoints, which are an additional source of transient perturbation at the scale of the catchment. Therefore, in order to focus on perturbations associated with drainage divide

dynamics, basins that contain knickpoints are ignored. Our results highlight the major control of basin size on the dispersion $E/U$. When all classes of drainage areas are combined together, we still obtain a clear relationship between $\Delta\chi$ and $\Delta G$ and $E/U$ (Fig 6d and e). One may note however two different trends for victim and aggressor basins. Aggressors show a more scattered distribution for $\Delta\chi$ and $\Delta G$ metrics. When compared to victims, these basins have hillslopes closer to the critical value $Sc$ (Fig. S3). Hence, the dispersion may be explained by the non-linear relationship existing between denudation rates

and basin slope (Montgomery and Brandon, 2002; Binnie et al., 2015). Therefore, a simple linear trend is no longer sufficient to properly fit our results. Hence we consider victim and aggressor basins independently assuming two linear trends between $E/U$ and each aggressivity metric.

## 4 Discussion

### 4.1 Sensitivity tests

The reference model involves various parameters related to uplift, hillslope denudation and fluvial incision. A systematic analysis of all parameters is out of the scope of this manuscript. Here, we do not investigate the effect of erodibility, because it remains poorly constrained in nature and because our study is mostly focused on channel head properties. In this section, we assess the sensitivity of the results to both tectonic and hillslope processes, by studying the specific impact of uplift $U$, diffusivity $D$ and critical hillslope gradient $Sc$.

### 4.1.1 Sensitivity to uplift rate

We vary the tectonic uplift rate from 0.5 mm.yr⁻¹ to 2 mm.yr⁻¹, which is in the range observed in moderately active mountain belts; e.g. the Alborz, the Alps or the Caucasus (Champagnac et al., 2009; Djamour et al., 2010; Vincent et al., 2011). It is well-known that a river responds to a fall in base level (due to changes in rock uplift rate or other forcing) by cutting downward into its bed, deepening and widening its active channel. In our simulations, changes in uplift rate lead to

variations in the geometry of the drainage network. Compared to the reference model, an uplift rate of 2 mm.yr⁻¹ (0.5 mm.yr⁻

Earth **Surface**
**Dynamics**
Discussions

[1]) results in an increase (decrease) of river channelization - inversely proportional to drainage density - which induces larger (smaller) river basins, a lower (higher) range of values for $\Delta\chi$ and $\Delta G$ and a higher (lower) range of values for $\Delta H$ (Fig. 7). More importantly, our simulations suggest a linear relationship between tectonic uplift and denudation rates. As a result, assuming no climate-tectonic feedback we obtain no significant changes in the relationship between the calculated

aggressivity metrics and the $E/U$ ratio for uplift rates ranging between 0.5 mm.yr$^{-1}$ and 2 mm.yr$^{-1}$.

### 4.1.2 Influence of hillslope processes

Hillslope denudation is proportional to the diffusivity coefficient $D$ and depends on the critical slope Sc (Eq. 2). To test the effect of hillslope processes, we first vary $D$ between $10^{-3}$ and $10^{-1}$ m$^2$.yr$^{-1}$. We find no major difference with the reference model over that range of values for D (Fig. 8). Denudation intensity also varies inversely to the square of the critical

hillslope gradient $Sc$ (Eq. 2). Assuming a critical slope between 20° and 40°, we find that $Sc$ is a parameter influencing aggressivity metrics (Fig. 9). As in the case of variations in uplift rate, changes in $Sc$ lead to differences in the organization of the drainage network. Compared to the reference model, a low $Sc$ leads to large river channelization, reducing the range of all aggressivity metrics, but does not affect significantly the relationship between the $E/U$ ratio and the studied metrics. These observations are consistent with a landscape where hillslopes are completely determined by the assumed critical slope

value.

Altogether, these sensitivity tests demonstrate the robustness of our findings: $\Delta\chi$ and $\Delta G$ are, to the first-order, reliable metrics to assess the effect of divide mobility on basin-wide denudation rates inferred from simulations. In the following section, we apply this approach to field observations and discuss the consequences for sampling and interpretation.

### 4.2 Implication for basin-wide denudation rate interpretation

Over the last decades, measurements of cosmogenic radionuclides (CRN) concentration in alluvial sediments (see Granger et al., 2013 and references therein), of suspended sediments (Gabet et al., 2008) and of detrital thermochronology (Huntington and Hodges, 2006) have become common practices to assess basin-wide denudation rates. However, their interpretation remains debated, even in settings where topographic steady state is supposedly achieved regionally.

### 4.2.1 Application to the Great Smoky Mountains

As previously mentioned (Matmon et al., 2003a,b), while the Great Smoky Mountains in the southern Appalachians are expected to be in a quasi-topographic steady state, basin-wide denudation rates show a strong dispersion up to a factor of 2 in comparison with the estimated uplift rate (0.025 to 0.03 mm.yr$^{-1}$, see Fig. 1). Basins in the Great Smoky Mountains exhibit average slope values ranging between 20 and 30° (Matmon et al., 2003a). This may suggest that hillslopes are close to the

critical angle of repose. On the basis of these assumptions, we consider that our approach is applicable to the basins studied



by Matmon et al. (2003a,b), even though estimated uplift rates differ by nearly two orders of magnitude when compared to the uplift rates used in our models. We use the data associated with 40 basins originally sampled by Matmon et al., (2003a) and for which denudation rates were re-calculated by Portenga and Bierman (2011). Following our method, we calculate the three basin-averaged aggressivity metrics $\Delta\chi$, $\Delta G$ and $\Delta H$ associated with these 40 catchments (Fig. 10; See also Fig. S4).

The calculation of $\chi$ requires to define the elevation of the catchment outlets $H_b$ and the $m/n$ ratio (Eq. 4). As underlined by Forte and Whipple (2018), the choice of the "correct" outlet elevation is non-trivial in natural settings. We first consider a local base level given by the Tennessee river. To test the relevance of this choice, we also test a base level located at a fixed elevation $H_b$= 400 m. We use the same $m/n$ ratio value of 0.45 as used in the Willett et al. (2014) study for the Great Smoky Mountains. For all calculated metrics, the majority (ca. 75% for $\Delta G$ and 55% for $\Delta H$) of the basins are located in the

expected quadrants. However, more attention must be given to the results based on $\Delta\chi$. For this metric, ~62% of the analyzed basins lie in the expected quadrant when we consider the Tennessee river as the local base level versus ~67% for $H_b$= 400 m (Fig. 10). If the general result remains similar here, we show that the choice of a different base level $H_b$ leads to significant variations in $\Delta\chi$ for individual basins. This highlights the main weakness of the $\Delta\chi$ metric, which is highly sensitive to the choice of the proper base level $H_b$. Nevertheless, our results confirm the findings of Willett et al. (2014),

suggesting that a significant part of the data variance observed in the Matmon et al. (2003a) can be explained by divide migration (Fig. 10), raising this possible explanation for the variability of most natural data sets. One may note that the Southern Appalachians exhibit migrating knickpoints that can locally affect denudation rates (Gallen et al., 2011 ; Gallen et al., 2013). This last point can also explain part of the observed variability in this dataset but this specific impact is beyond the scope of the present manuscript.


Based on both our simulations and this field dataset, we propose to favor the use of the Gilbert metric $\Delta G$ based on the cross-divide contrast in channel head local slope (Forte and Whipple, 2018). Among the tested metrics, $\Delta H$ appears the least sensitive to disequilibrium, and $\Delta\chi$ requires better constraining and defining objective criteria for $H_b$.

### 4.3.2 Assessment of topographic disequilibrium

Topographic steady-state is a very convenient assumption and concept to deduce the uplift pattern in mountains ranges from denudation rates, and thus to obtain significant information on the geometry of active structures and on orogen dynamics (Lavé and Avouac, 2001, Godard et al., 2014 ; Scherler et al., 2014; Le Roux-Mallouf et al., 2015). However, this assumption is seldom verified at the scale of sampled watersheds.

On the basis of our modeling, we show that the competition between low-order basins has a significant impact on basin-wide

denudation rates. The proposed approach provides a new tool to assess the potential maximum deviation from topographic steady state based on aggressivity metrics and drainage area, which can both be inferred from a simple DEM. The closer to



zero the aggressivity metrics, the more representative to uplift rate are the measured denudation rates. However it still remains dispersion of denudation rates values, especially for smaller catchments.

Based on our sensitivity tests for moderately active orogens (with uplift rates between 0.5 and 2 mm/yr), the empirical relationship between $\Delta G$ and $E/U$ obtained from the reference model (Fig. 6) can be used to assess the topographic disequilibrium of basins. Especially for victim basins ($\Delta G < 0$), this relationship exhibits a linear relationship:

$$\frac{E-U}{U} = 0.03 \Delta G \qquad \text{for basin area} > 50 \, \text{km}^2 \, , \tag{6}$$

For the sake of simplicity our models involve spatially homogenous and time invariant parameters. Additional simulations are now needed to test this approach in more complex settings, including spatial and temporal variability in climate and tectonic forcing or internal landscape parameters like erodibility.

### 4.4.3 Improvement of sampling strategy

Basin-wide denudation rates obtained from CRN concentration measurements, suspended sediments or detrital thermochronology depend on many parameters including lithology, ice cover, rainfall, landslide activity or tectonic uplift (Vance et al., 2003; Bierman and Nichols, 2004; Wittmann et al., 2007; Yanites et al., 2009; Norton et al., 2010; Godard et al., 2012; Whipp and Elhers, 2019). Hence, to unravel the influence of tectonics from other processes, a specific sampling strategy is usually recommended: (1) to sample catchments with homogeneous lithologies to limit the effect of spatial variations in the abundance of target minerals in bedrock formations; (2) to select catchments with no ice cover (past or present) because the input of glacier-derived sediments can significantly complicate the interpretation of CRN concentrations; (3) to choose areas with spatially uniform rainfall distribution; and (4) to consider watersheds where the relative contribution of landslides to long-term landscape evolution is low. Unfortunately, these different criteria imply to select watersheds with variable sizes. The first three criteria favor the sampling of small catchments, whereas the last one requires basins large enough to be less affected by landslides. Indeed, Niemi et al. (2005) proposed that mixing effects efficiently dampen the stochastic nature of hillslope sediment delivery by landsliding above a critical catchment area. Considering an uplift rate of 0.5-2 mm.yr[-1], the recommended minimum area needed to mitigate these biases associated with the stochastic input from landslides is of 50 to 200 km[2].

Based on our simulations, a relationship between maximum of erosion variability (0.5 and 99.5 percentiles, respectively) due to divide mobility $[(E-U)/U]_{max}$ and the catchment size $A$ can be derived (Fig. 11). Our results suggest a logarithm dependence between these two parameters, regardless of the assumed $U$, $D$ and $S_c$ :

$$[(E-U)/U]_{max} = c_1 \log(A) + c_2 \quad \text{for} \quad 1 \, \text{km}^2 < A < 100 \, \text{km}^2 \, , \tag{7}$$

For victim basins ($\Delta G < 0$), $c_1 \simeq 0.05$ and $c_2 \simeq -0.5$, whereas for aggressor basins $c_1 \simeq -0.14$ and $c_2 \simeq 1$. This provides
a new additional guideline for the design of sampling strategies in terms of basin size. For instance, considering a quasi-steady state mountain belt with an uplift rate of 1 mm.yr$^{-1}$, the minimum basin area required for an erosion rate variability lower than 0.5 mm.yr$^{-1}$ is ca. 1 km$^2$ for victim basins and 30-40 km$^2$ for aggressor basins.

## 5 Conclusions

Calculations from a Landscape Evolution Model assuming spatially uniform uplift, rock strength and rainfall confirm that
the concept of topographic steady state is relevant at the scale of entire mountain belts, but represents an oversimplification at the scale of individual watersheds. Our simulations underline the role of divide mobility on deviations from equilibrium, which can lead to significant differences between tectonic uplift rate and basin-wide denudation rates even if an overall topographic steady state is achieved at large scale.

To better assess these deviations, we propose new basin-averaged aggressivity metrics based on the approach of Willett et al.
(2014) and Forte and Whipple (2018). They include mean cross-divide contrasts in channel-heads $\chi$, local slope and elevation at the scale of entire river basins. From our calculations, contrasts in channel-head elevation appear to be weakly sensitive to local disequilibrium, whereas the basin denudation-to-uplift ratio $E/U$ exhibits a nonlinear relationship with $\Delta \chi$ and $\Delta G$. Together, these two metrics reveal that $E/U$ depends on both basin aggressivity and basin area. This last parameter has a key control on the dispersion in $E/U$, which can reach a factor of two, regardless of the imposed uplift rate (here 0.5-2
mm.yr$^{-1}$), diffusivity (here $10^{-3}$-$10^{-1}$ m$^2$.yr$^{-1}$) and hillslope gradient (here 20°-40°). By comparing our results to CRN measurements from the Great Smoky Mountains (Matmon, 2003a,b), we show that this approach can be used to improve field sampling strategies and provides a new tool to derive a minimal uncertainty in basin-wide denudation rates due to topographic disequilibrium.

## Acknowledgments

Timothée Sassolas-Serrayet's Ph.D. is supported by a fellowship from the French Ministry for Higher Education. We thank Wolfgang Schwanghart and Benjamin Campforts for providing the programme TopoToolBox and TTLEM to analyze and to simulate landscape evolution. We acknowledge funding of the French Agence National de la Recherche, grant ANR-18-CE01-0017 (Topo-Extreme). This study contributes to the IdEx Université de Paris ANR-18-IDEX-0001. This is IPGP contribution # XXX.



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



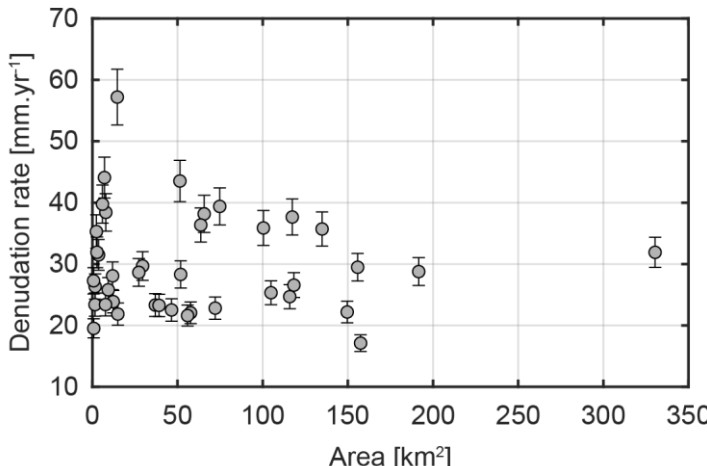

**Figure 1: Basin-wide denudation rate variability as a function of drainage area in the Great Smoky Mountains. Data originally from Matmon et al., 2003a, denudation rates are reprocessed by Portenga and Bierman, (2011).**

525







**Figure 2: Map view of the temporal evolution of the reference model. Colorbar gives the model elevation. Black lines show the evolution - and the migration - over time of drainage divides for five drainage basins. One basin is colored in orange to underline its expansion. The red circle in figure 3d shows the location of an imminent drainage capture after 2.5 Myr of simulation (see Supplementary Video n°1).**





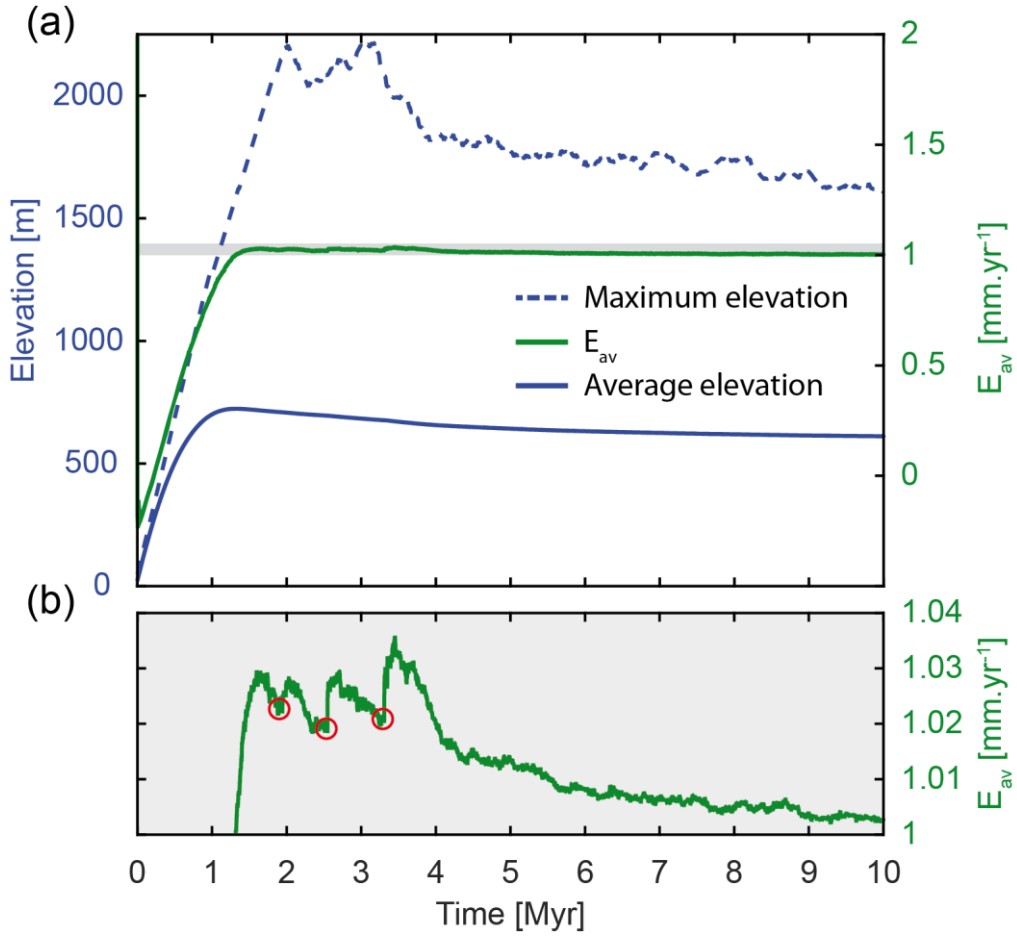

**Figure 3: Evolution of the reference model over time. (a) Average elevation (blue solid line), maximum elevation (blue dashed line) and average denudation rate (green solid line) over the whole model. (b) Expanded view of the mean denudation rate of figure 3a (in the light grey area). Red circles highlight significant stream captures that lead to an abrupt increase in average denudation rates over a subsequent period of several time steps.**



**Figure 4: Variability of denudation rates over time for a compilation of five simulations of the reference model. (a) to (c) Variability of denudation rate as a function of basin area after 2.5, 5 and 10 Myr of simulation, respectively. (d) Mean absolute deviation (MAD) from uplift rate (1 mm.yr⁻¹) for three sets of basin sizes: 1-2 km², 10-20 km² and 100-200 km². Negative (positive) deviation is related to basins with a deficit (excess) of denudation. The orange color gradient corresponds to the transient period associated with the rising plateau before the model reaches an average topographic equilibrium.**





550

**Figure 5: Denudation rates and cross-divide contrast metrics obtained for the reference model after 5 Myr. (a) Map of denudation rates for basins of 2-4 km². Black thick lines correspond to basin divides in figure 2e. Black arrows, show the direction of divide migrations for one basin. (b) χ map. (c) Channel slope map. (d) Channel elevation map.**





**Figure 6: Denudation rates normalized by uplift rate as a function of aggressivity metrics. Note that the x axis is reversed for both $\Delta\chi$ and $\Delta H$. Color scale indicates basin area. (a) to (c) Basins of 10-20 km² for reference model after 2.5 Myr. Red squares correspond to basins that contain at least one knickpoint. (d) to (f) Basins of variable sizes, classified in seven area classes from 1 to 128 km², over the time period 2.5-10 Myr. Red lines are linear fits for victim and aggressor basins.**

Earth **Surface**
**Dynamics**
Discussions
EGU
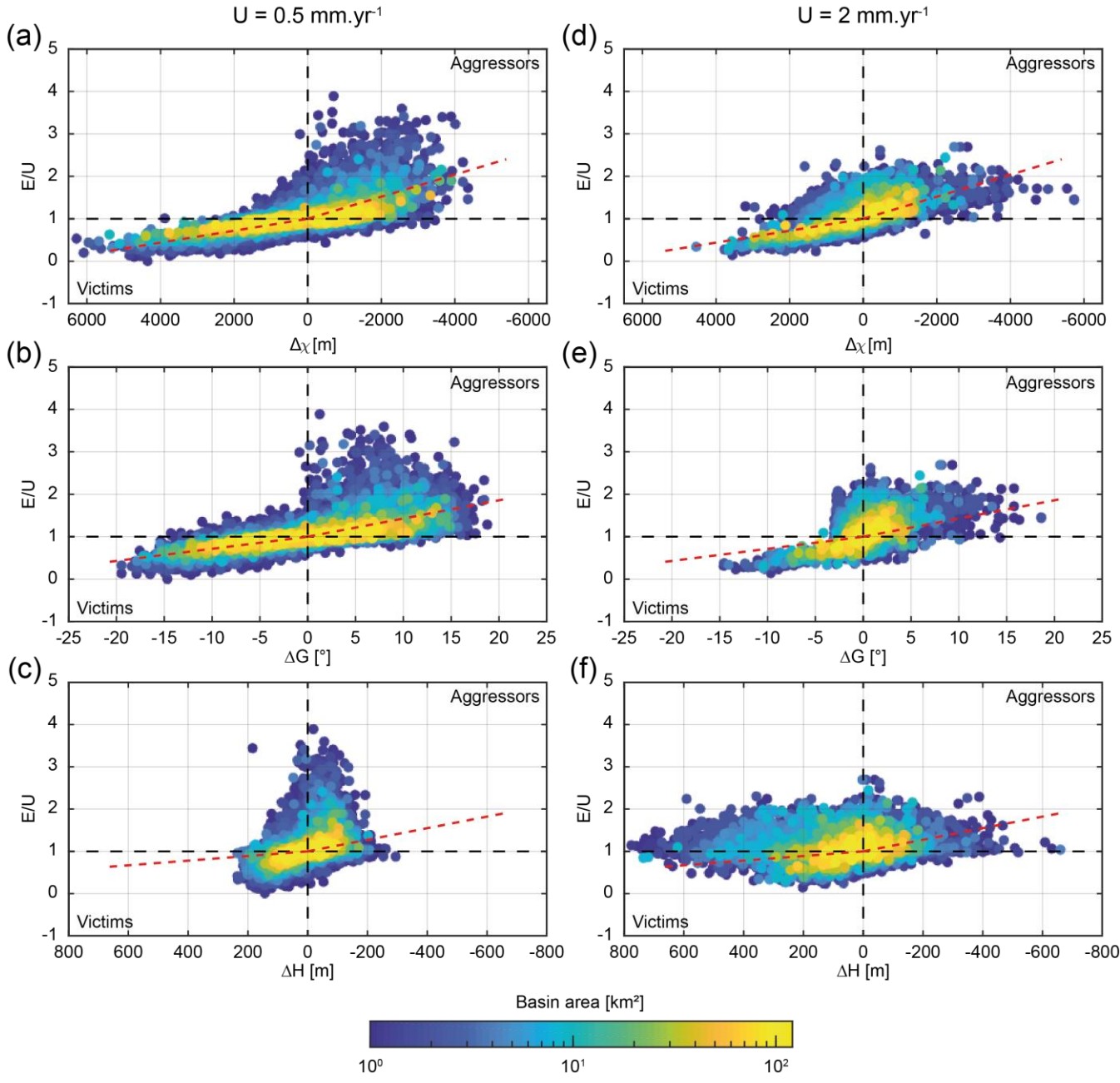

**Figure 7: Sensitivity to uplift rates.** Note that the horizontal axis is reversed for both $\Delta\chi$ and $\Delta H$. Color scale indicates basin area.
565 Red dashed lines correspond to the linear fits obtained for the reference model (see figure 6). (a) to (c) Same as Fig. 6d-f, with an uplift rate of 0.5 mm.yr$^{-1}$. (d) to (f) Same as Fig. 6d-f, with an uplift rate of 2 mm.yr$^{-1}$.



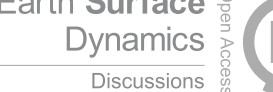

Figure 8: Effect of diffusivity. Note that the x axis is reversed for both $\Delta\chi$ and $\Delta H$. Color scale indicates basin area. Red dashed lines correspond to the linear fits obtained for the reference model (see figure 6). (a) to (c) Same as Fig. 6d-f, with a diffusivity of $10^{-3}$ m$^2$.yr$^{-1}$ . (d) to (f) Same as Fig. 6d-f, with a diffusivity of $10^{-1}$ m$^2$.yr$^{-1}$.







575 **Figure 9: Effect of critical slope. Note that the x axis is reversed for both $\Delta\chi$ and $\Delta H$. Color scale indicates basin area. Red dashed lines correspond to the linear fits obtained for the reference model (see figure 6). (a) to (c) Same as Fig. 6d-f, with a critical slope of 20° . (d) to (f) Same as Fig. 6d-f, with a critical slope of 40°.**





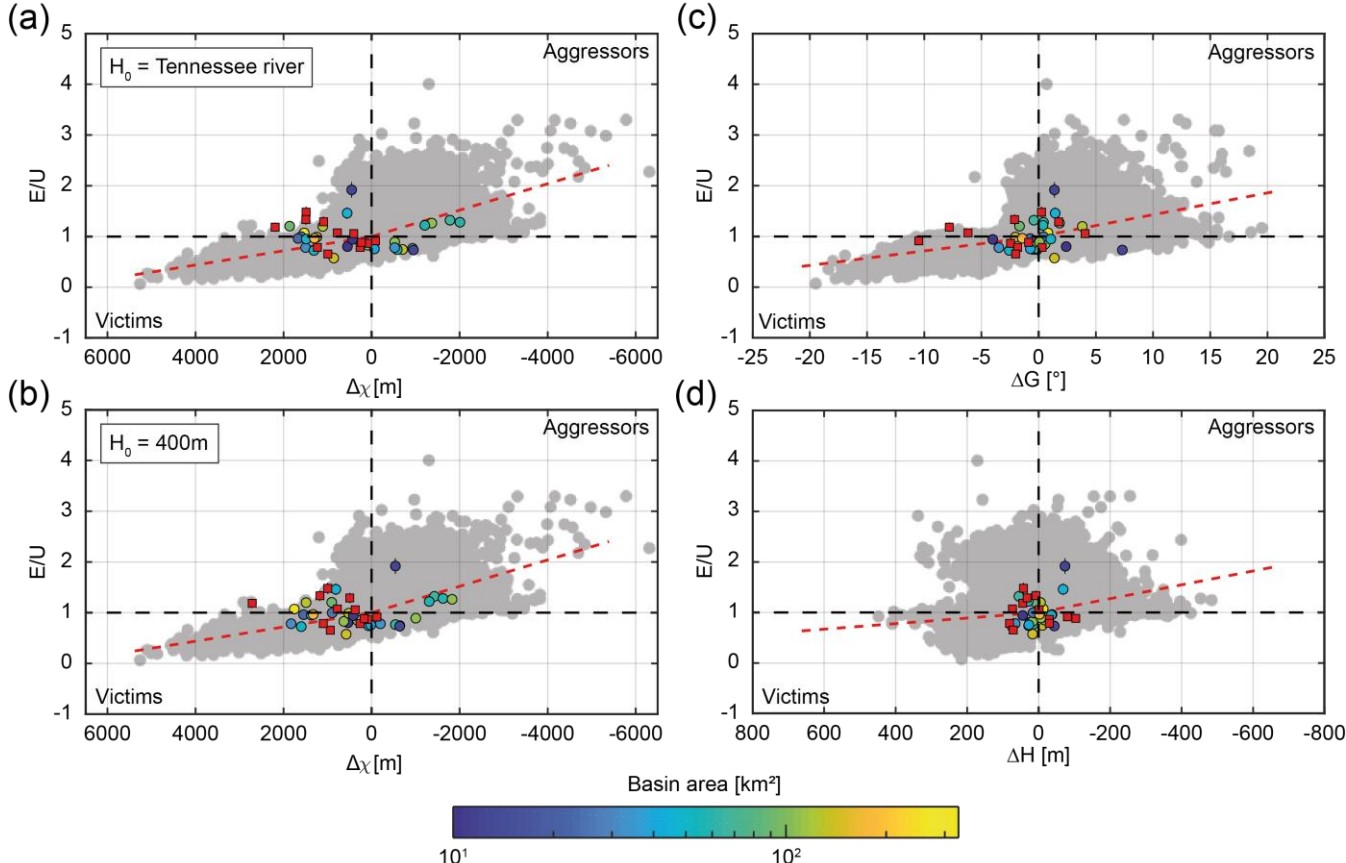

 **Figure 10: Denudation rates in the Great Smoky Mountains as a function of aggressivity metrics. Note that the x axis is reversed for both $\Delta\chi$ and $\Delta H$. Color circles are reported according to the $E/U$ ratio for basins sampled by Matmon et al., 2003b, with $E$ recalculated by Portenga et al., 2011, error bars are represented with thin black line but are almost always smaller than symbols. Color scale of these circles is related to basin size. Red squares are basins with an area less than 10 km². Red dashed lines correspond to linear fits obtained for the reference model. Grey circles in the background correspond to the reference model (see**
 **figure 6). (a) and (b) Relationship between denudation rates and $\Delta\chi$ with a base level corresponding to the Tennessee river and at a fixed elevation of 400 m, respectively. (c) and (d) Relationship between denudation rates and Gilbert aggressivity metrics, $\Delta G$ and $\Delta H$, respectively.**



Earth **Surface**
**Dynamics**
Discussions
EGU

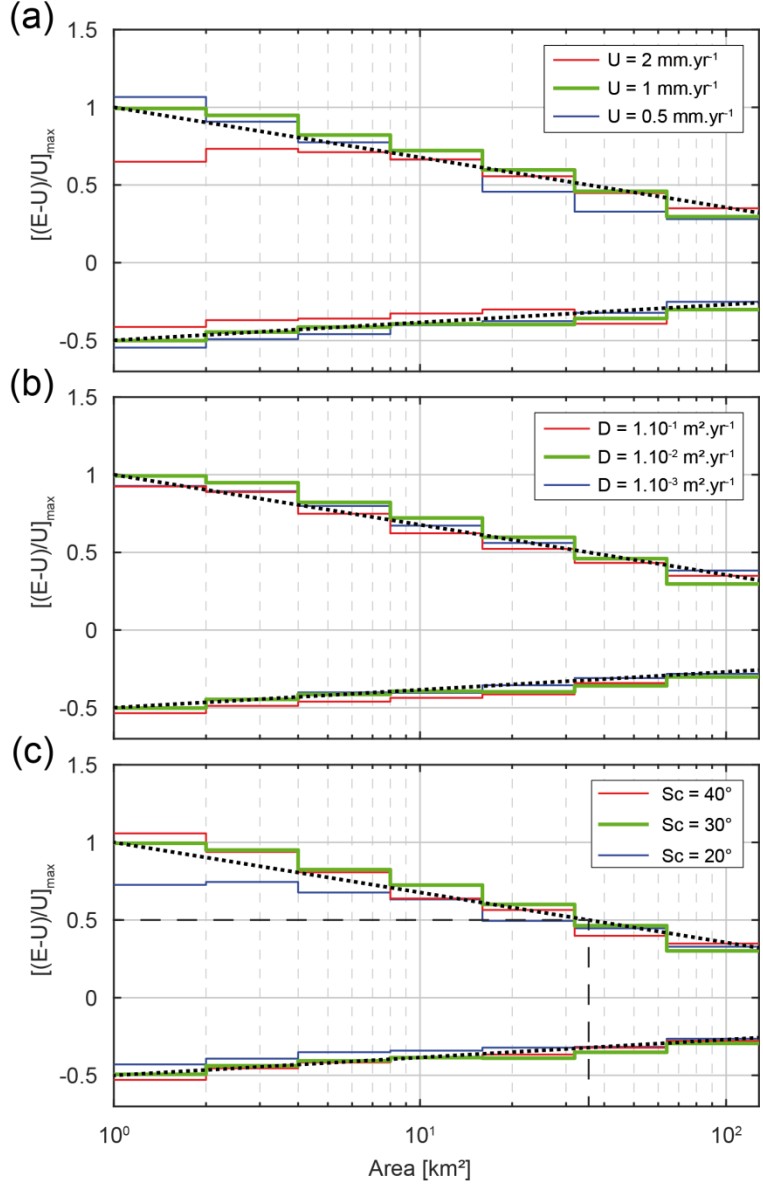

590

**Figure 11: Deviation from steady state due to drainage migration as a function of basin size. Color lines are related to the maximum dispersion of denudation rates (0.5 and 99.5 percentiles) due to divide mobility. Green lines are associated with the results obtained for the reference model. Dotted black lines represent the logarithm relationship between $[(E - U)/U]_{max}$ and the basin area for victim and aggressor watersheds (Eq.7). Seven sets of basin size are considered: 1-2, 2-4, 4-8, 8-16, 16-32, 32-64 and**
595 **64-128 km² every 0.5 Myr between 2.5 and 10 Myr. (a) Effect of uplift rate. (b) Effect of diffusivity. (c) Effect of critical slope. Thin dashed black lines show the minimal basin area required for a maximal denudation rate deviation from uplift rate equal to 0.5.**