# Peer review of "Estimating the disequilibrium in denudation rates due to divide migration at the scale of river basins."

_Earth Surface Dynamics, 2019_

## Referee Comment (RC2)

**Review of 'Estimating the disequilibrium in denudation rates due to divide migration at the scale of river basins' by Sassolas-Serrayet et al.**

**Fiona Clubb**

**August 12, 2019**

The paper in review presents a series of numerical models set up to investigate the impact of drainage divide migration on landscape denudation rates, as well as both presenting new topographic metrics and testing the ability of previous ones to detect divide migration. The authors then apply these metrics to provide some useful constraints for basin size for CRN sampling. I think the paper is interesting, well written, and builds well on previous work that quantifies the extent of drainage divide migration across landscapes, given the many papers that have been published in the last few years on the topic. However I have some concerns about the setup of the model, in particular the use of a parametrised critical area threshold. I therefore think it will be very suitable for publication in ESURF provided that the comments detailed below can be addressed.

Line 34 - 35: repetition of 'modern landscapes' twice in the sentence.

Eq 1: I think the fluvial expression should be $A > A_c$?

Eq 1: Similar to Reviewer 1, I agree that the model setup described in equation 1 makes it seem like the position of channel heads and resulting drainage divide metrics will be determined by the critical area threshold ($A_c$) parameter. I'm not convinced that, in real landscapes, there is a critical area threshold for determining channel head location or, if there is, that this should be fixed across the landscape as a whole. This setup seems a bit unintuitive to me - why not just combine fluvial incision with hillslope diffusion at each node and eliminate the need for an $A_c$ parameter in the LEM at all?

Line 118: Much work has shown that in many real landscapes it is most likely that $n \neq 1$ [e.g. *Lague*, 2014; *Harel et al.*, 2016]. This will again have a significant impact on the distribution of slopes and erosion rates within the model landscapes, and may influence the calculation of the divide migration metrics. I think it would be useful to run some test landscapes where $n > 1$ to determine i) the impact this has on the variability of erosion rate with basin area; and ii) the impact on the aggressivity metrics.

Line 121: Following on from this, I would suggest running a sensitivity analysis changing $A_c$ in the model setup to see what effect this has on the calculation of divide migration metrics.

Line 125: How is it determined whether the model runs have indeed reached steady state?

Line 135: It's a bit unclear how the elevation is calculated. Is this the average elevation for all pixels in the basin at each time step?

Line 162: It would be good to include some more details here of how this averaging is carried out. Is local slope calculated using a moving window (including hillslopes), or is this just the slope of the first order channel? It wasn't clear to me whether the elevation at each channel head was averaged, or whether the first order channel downstream of the channel head was included.

Fig S2: I think it would be useful to edit this figure and move it to the main paper, as it was difficult to follow how the aggressivity metric is calculated from the text. I realise some of this is based on previous work, but the averaging of the cross-divide metrics to produce a new metric for each basin is novel in this paper. I didn't understand how the segmenting shown in Fig S2 was done, or how the segment length over which the averaging should be performed is determined.

Line 221: As well as drainage divide migration, variability in erosion rate between basins could simply result from the transient propagation of knickpoints, especially in the earlier model runs. This seems to be supported by the fact that the variability decreases significantly through time as shown in Figures 4 (a) - (c).

What is the evidence that this variability is in fact due to divide migration and not due to transient knickpoint propagation? In the text it's stated that this is shown from Fig 2(e), but I didn't understand how this figure shows that.

Figure 6 and 7, and Lines 259 - 262: Is there any physical meaning/theoretical prediction for the linear trends on these figures, and how significant are they? If indeed there is a non-linear relationship between S and E, then I wouldn't expect a linear relationship between $\Delta G$ or $\Delta H$ and E/U.

Figure 7: I was quite surprised how noisy some of these data are, especially for the smaller basins, considering that these are all from LEMs and not real landscapes. Maybe this could do with a bit more discussion in the text as to potential reasons for this noise? In the text it's stated that it is due to the presence of significant knickpoints, but I was confused as to whether basins with knickpoints were excluded or not. It raises the possibility that some of these metrics, especially $\Delta H$, would be too noisy in real landscapes when additional factors such as variations in lithology, rainfall or uplift are taken into account.

Lines 275 - 277: I'm also surprised that there is an increase in drainage density with increasing uplift when $n = 1$ in the model runs. Previous theoretical work by *Tucker and Bras* [1998] predicted that drainage density should be independent of erosion rate when $n = 1$, which was then shown by numerical modelling using CHILD by *Clubb et al.* [2016] (Figures 4 and 5). Furthermore, and more qualitatively, when I have run LEMs with detachment-limited stream power in the past I have generally found that the geometry of the network remains fixed when increasing uplift rate, and only the slopes of the network increase. I think this could be discussed in more detail as to why there is this discrepancy with previous theoretical predictions and numerical modelling results.

Line 279: 'we obtain no significant changes in the relationship between the calculated aggressivity metrics and the E/U ratio for uplift rates...' I'm not sure I agree with this statement - from Fig 7, some of the distributions look quite different for U = 0.5 and U = 2 mm/yr, especially for $\Delta H$. This may be just because of the smaller basins, or that it's difficult to get a sense of how dense the data are in the centre of the plot.

Lines 276 and 287: I think it would strengthen the paper to quantify this change in 'river channelization' proposed by the authors. At the moment this appears to be a qualitative statement which is difficult to verify from the current figures and analysis. It's an interesting result that changing the value of $S_c$ influences drainage density, which makes sense from the theory and has implications for real landscapes where $S_c$ is difficult to determine. I think more could be made of this, and suggest simply calculating drainage density for the different model runs. This would put some more weight behind the statement that increasing drainage density is the mechanism by which changing $U$ and $S_c$ impact the aggressivity metrics.

Lines 314 - 315: I don't understand what are the 'expected quadrants' that the basins are being compared to here. Is this compared to the reference model, or compared to Willett et al. (2014)?

**References**

Clubb, F. J., S. M. Mudd, M. Attal, D. T. Milodowski, and S. W. Grieve, The relationship between drainage density, erosion rate, and hilltop curvature: Implications for sediment transport processes, *Journal of Geophysical Research: Earth Surface*, p. 2015JF003747, 2016.

Harel, M. A., S. M. Mudd, and M. Attal, Global analysis of the stream power law parameters based on worldwide 10be denudation rates, *Geomorphology*, *268*, 184–196, 2016.

Lague, D., The stream power river incision model: evidence, theory and beyond, *Earth Surface Processes and Landforms*, *39*, 38–61, 2014.

Tucker, G. E., and R. L. Bras, Hillslope processes, drainage density, and landscape morphology, *Water Resources Research*, *34*, 2751–2764, 1998.

---

## Referee Comment (RC1) · Adam Forte (Referee) · 24 Jul 2019

I've completed my review of Sassolas-Serrayet et al's paper entitled 'Estimating the disequilibrium in denudation rates due to divide migration at the scale of river basins.' In this manuscript, the authors analyze a series of landscape evolution models to try to ultimately assess the ability for one to measure the background rock uplift rate through basin averaged erosion rates in the presence of divide migration and find that there is a drainage area dependence on the ability to do so. Using this, they are able to provide some recommendations for planning of sampling for things like cosmogenic erosion rates. This study seems timely and interesting and is a good fit for Earth Surface Dynamics. While I think the general results are strong and consistent with what others have shown (or rather what is implied by the results others have shown), I do have some concerns about whether they've biased their detailed results with the specifics of their model setups. Much hinges on whether or not the models were ran with an imposed threshold area or not (it is a little ambiguous in the text, so it's possible that my concerns are for naught and I simply misunderstood what they meant). These concerns (along with other comments that the authors hopefully find helpful) are outlined in my detailed comments below.

L37-38: Probably important to clarify that the divides are migrating in response to the same change (it's implied, but not explicitly stated in the way you word it).

Equation 1: I think the top expression (U + (dz/dt)_fluv) should be for A > A sub c, correct?

L119 – 121: Is the stated critical drainage area applied in the model (i.e. the model is run with the rule set such that diffusion is only applied where A<A sub c and incision is only applied where A>A sub c) or is this simply the threshold area used for extracting the channels (and thus the channel heads) for analysis? My (anecdotal) experience has been that running TTLEM (or really any model that allows you to do so) with an explicit critical drainage area that is built in where it defines where incision/diffusion is applied can produce some odd behaviors and very odd drainage networks. If you were running TTLEM with the critical drainage area option turned on, did you experiment with the sensitivity of your results to turning this off (i.e. setting it to 0)? It's important to note that this option is kind of atypical, i.e. it is not something allowed in CHILD, LandLab, etc. I think more importantly, if you are running with Ac (or AreaThresh as it's named in the TTLEM setup) set to a value greater than zero you are artificially controlling the length of the hillslope (it would otherwise be set by the combination of K and D values you provide) and thus controlling the length scale over which the landscape responds to the divide migration (as this is happening mostly in the hillslopes based on prior results). This may in turn reflect some of your other results (e.g. erosion rates as a

function of drainage area, etc).

L125-126: This is the theoretical time to steady-state following a perturbation and assuming a fixed drainage area, which is not really applicable to the time to steady-state for a model ramping up (i.e. running from an initial random noise, low elevation topography to a stabilized topography). I don't think this necessarily matters that much to your results as you are basically just exploiting the fact that this portion of a model run has a lot of drainage reorganization, but I would be careful about equating these.

L205-207: This pattern in erosion rates as a function of drainage area pretty much follows from the observation discussed in Forte & Whipple 2018, namely that if considering simulated landscapes experiencing progressive divide motion (as opposed to discrete captures), the erosion rate contrasts across divides is very spatially limited to areas very near the divides (essentially hillslopes), thus as you move to larger drainage areas, the ability to 'see' this across divide contrast in erosion rate in basin averaged values would be expected to decrease as the signal is diluted by more and more of the drainage basin eroding very near the background erosion rate / uplift rate.

L232-236: This all seems logical, however it might be important to note that basin averaged statistics like this work best when a basin is either uniformly (or at least consistently) either expanding or contracting. This is probably (usually) the case in homogeneous models like the ones you use here, but in either more heterogeneous models or when applied to real landscapes, there is the danger of a basin appearing to be neutral because it is expanding in one direction and contracting in another (i.e. the metrics counterbalance when averaged over the whole basin). This could be especially noticeable when divide migration is driven by a lateral gradient in uplift rate with respect to the main drainage direction. Thus, I would (maybe somewhat self importantly) argue that along-divide metrics are still quite useful to consider.

L249: This is also consistent with what Forte & Whipple 2018 saw in natural landscapes, i.e. across divide contrasts in elevation were usually equivocal in terms of

indicating potential for drainage divide motion compared to the other metrics.

L254-255: Why do any of these basins have knickpoints? You're applying a constant uplift rate and progressive divide motion shouldn't really impart knickpoints onto any profiles. Are they coming from captures? While I understand the logic of ignoring basins with knickpoints, it's more that I wonder if the presence of knickpoints in this is suggesting that there may be some stability issues. One of the behaviors of the TVD-FVM algorithm is to keep and accentuate any knickpoint (even if those are developed through numerical instability), so it would be good to try to diagnose why there are knickpoints in the first place to rule out model instability. You could try running one of your models with the same exact setup but using the implicit (fastscape) algorithm that's built into TTLEM and see if you also are getting knickpoints.

L274-278: I'm a bit confused by this statement. From my own simulations with TTLEM, if you're starting with the same random noise and keeping everything else the same (i.e. not changing the length of timestep, etc), the drainage network evolution will be pretty much the same regardless of uplift rate. I would expect changes in the effective drainage density to manifest more in response to changes in the diffusion constant or ratios between K and D.

L283-285: Again, this might be a result of setting the area threshold to a non zero value (assuming you did). I'm also not sure how to interpret Sc if the area threshold is set to a non zero value, because you're artificially controlling the hillslope length and thus (I think) artificially controlling the maximum slope that can develop anyways. If I misunderstood your discussion of A sub c earlier and you were not running with A sub c set to a non zero value, feel free to ignore this comment.

L340-345: This is neat, though (and as you mention in the following sentence) it would be interesting to think about the applicability of this in more heterogeneous environments where, as I mentioned before, there's a greater chance that the basin averaged metric could be misleading (i.e. a low average metric because of coexisting drainage

**ESurfD**

Interactive
comment

area loss and gain on different sides). Maybe a valuable approach to consider would be including the standard deviation (or min and max) in the basin averaged metric to try to capture this potential variability without having to look at the divide segments in detail?

L349-372: This is a good thing to focus on, but I think you could add an interesting discussion here of considering the sampling strategy with regards to the goal of the study. At present, this gives a (valuable) set of ideas for how large a basin needs to be to get an accurate assessment of the uplift rate from the erosion rate, but alternatively you could point out that this gives you a sense of the size of basin you need to target if you're explicitly interested in measuring divide migration rates, i.e. larger basins are not going to be helpful. Similarly, this speaks to the need to pre assess basins for their potential divide mobility before sampling (if your intention is to get at background uplift rate), i.e. if either the divide segment or basin average metrics suggest no divide mobility, you don't need to worry about the size of the basins as much (except for all the other concerns we already have, that you mention).

---

## Short Comment (SC1) · 15 Aug 2019

I enjoyed reading this manuscript and appreciate the strong effort at quantifying intra-basin variability under regional quasi-steady state conditions. It's an important problem and I think this is a very good approach with interesting results. I just wanted to draw the authors' attention to our paper from last year which they might find relevant. We quantified divide migration from event-triggered landslides and found that, although divides generally moved in directions predicted by cross-divide gradients in the Gilbert metrics (which we have attempted to place in the context of progress toward regional steady-state in Taiwan), landslide stochasticity introduces a lot of complications that

[Figure]

**ESurfD**
* * *
Interactive
comment

would be especially pronounced in small basins. Additionally, we had similar struggles with the use of $\Delta\chi$, finding that it works well at predicting divide migration where identifying base level is very simple and poorly where it is not. I thought this was relevant to mention, as many of our conclusions drawn from cross-divide observations at the timescale of a single earthquake/storm agree with the authors' long timescale basin-scale observations.

Here is the paper in question:

Dahlquist, M. P., West, A. J., and Li, G., 2018, Landslide-driven drainage divide migration: Geology, v. 46, no. 5, p. 403–406, doi:10.1130/G39916.1.

---

## Author Comment (AC2) · 4 Oct 2019

RC2: The paper in review presents a series of numerical models set up to investigate the impact of drainage divide migration on landscape denudation rates, as well as both presenting new topographic metrics and testing the ability of previous ones to detect divide migration. The authors then apply these metrics to provide some useful constraints for basin size for CRN sampling. I think the paper is interesting, well written, and builds well on previous work that quantifies the extent of drainage divide migration across landscapes, given the many papers that have been published in the last few years on the topic. However I have some concerns about the setup of the model, in

particular the use of a parametrised critical area threshold. I therefore think it will be very suitable for publication in ESURF provided that the comments detailed below can be addressed.

AC: Thank you for your positive and constructive comments. We carefully addressed every remark you formulated. Your concerns about threshold area Ac to distinguish hillslope and fluvial domains were justified. In the revised manuscript we produce a new set of results using simulations with no threshold area (Ac=0).This change largely corrects the atypical behaviors underline by both reviewers concerning the length of hillslope, the dependency of drainage network evolution to rock uplift, and the meaning of Sc, but it does not modify the major results concerning the impact of drainage migration on basin-wide denudation rates. We provide a revised version of the manuscript where we present our new results and addressing the referees comments.

RC2: Line 34 - 35: repetition of 'modern landscapes' twice in the sentence.

AC: We agree. Action: We reworded the sentence accordingly.

RC2: Eq 1: I think the fluvial expression should be A > Ac?

AC: We agree. Action: As modified our model with Ac = 0, we removed this condition in Equation 1. See RC1 for details.

RC2: Eq 1: Similar to Reviewer 1, I agree that the model setup described in equation 1 makes it seem like the position of channel heads and resulting drainage divide metrics will be determined by the critical area threshold (Ac) parameter. I'm not convinced that, in real landscapes, there is a critical area threshold for determining channel head location or, if there is, that this should be fixed across the landscape as a whole. This setup seems a bit unintuitive to me - why not just combine fluvial incision with hillslope diffusion at each node and eliminate the need for an Ac parameter in the LEM at all?

AC: We agree. See RC1's comment. Action: See RC1.

RC2: Line 118: Much work has shown that in many real landscapes it is most likely that

n > 1 [e.g. Lague, 2014; Harel et al., 2016]. This will again have a significant impact on the distribution of slopes and erosion rates within the model landscapes, and may influence the calculation of the divide migration metrics. I think it would be useful to run some test landscapes where n > 1 to determine i) the impact this has on the variability of erosion rate with basin area; and ii) the impact on the aggressivity metrics.

AC: We agree this would be a very interesting alley to explore. However, this would constitute an entirely new study that would go far beyond the scope of the present study. Action: We mentioned this as a perspective in the conclusions.

RC2: Line 121: Following on from this, I would suggest running a sensitivity analysis changing Ac in the model setup to see what effect this has on the calculation of divide migration metrics.

AC: We agree this could be a worthy path to follow. However, following comments from both referees we opted to set Ac = 0 in line with the majority of works published previously in order to produce results that can be compared.

RC2: Line 125: How is it determined whether the model runs have indeed reached steady state?

AC: As expected, the model does not actually reach a perfect steady state but a quasi-static equilibrium. First, we study the reorganization of the drainage after an initial perturbation keeping in mind that the drainage network shall not be affected by the initial model geometry. Next, in our simulations, we consider that the landscape reaches a regional steady state once transient topographies are completely eroded, i.e. once the mean model elevation remains constant and when the average denudation rate and the imposed rock uplift rates deviate by less than 2%. Finally, transient events such as discrete stream captures susceptible to form knickpoints are carefully detected and discarded from the results.

RC2: Line 135: It's a bit unclear how the elevation is calculated. Is this the average

elevation for all pixels in the basin at each time step?

AC: Basin-wide denudation rate is calculated using the average of differences of elevation in the basin borders during a 10 kyr interval.

RC2: Line 162: It would be good to include some more details here of how this averaging is carried out. Is local slope calculated using a moving window (including hillslopes), or is this just the slope of the first order channel? It wasn't clear to me whether the elevation at each channel head was averaged, or whether the first order channel downstream of the channel head was included. Fig S2: I think it would be useful to edit this figure and move it to the main paper, as it was difficult to follow how the aggressivity metric is calculated from the text. I realise some of this is based on previous work, but the averaging of the cross-divide metrics to produce a new metric for each basin is novel in this paper. I didn't understand how the segmenting shown in Fig S2 was done, or how the segment length over which the averaging should be performed is determined.

AC: We agree we did not provide enough detail on this. All metrics are measured at a reference drainage area. Across-divide metric differences are only calculated along divide segments shared by two reference basins. Hence, aggressivity metrics are the averaging of these documented segments along the perimeter of the sampled basin. Action: We reshaped section 2.3.2 and transferred Fig. S2 from the Supp. Mat. to the main text (now Fig. 2).

RC2: Line 221: As well as drainage divide migration, variability in erosion rate between basins could simply result from the transient propagation of knickpoints, especially in the earlier model runs. This seems to be supported by the fact that the variability decreases significantly through time as shown in Figures 4 (a) - (c). 1 What is the evidence that this variability is in fact due to divide migration and not due to transient knickpoint propagation? In the text it's stated that this is shown from Fig 2(e), but I didn't understand how this figure shows that.

AC: It is true that transient propagation of knickpoints can affect basin-wide denudation rates. To avoid any erosion signal associated with these transient features, we do not take into account the first stage of the model (< 2Myr for the reference model), when as you highlighted it, major knickpoints are retreating along the edge of a plateau. We also discard from the analysis all basins that contain knickpoints generally formed by discrete stream captures.

RC2: Figure 6 and 7, and Lines 259 - 262: Is there any physical meaning/theoretical prediction for the linear trends on these figures, and how significant are they? If indeed there is a non-linear relationship between S and E, then I wouldn't expect a linear relationship between _G or _H and E/U.

AC: We agree that the relationship between aggressivity metrics and E/U may be more complex than a simple linear relationship (especially when considering the different trends between aggressor and victim basins). We do not propose any physical meaning to explain this relationship, but we simply use it to compare results in regard to the different parameters. Action: We agree this may be misleading and we decide to remove this aspect from our analysis.

RC2: Figure 7: I was quite surprised how noisy some of these data are, especially for the smaller basins, considering that these are all from LEMs and not real landscapes. Maybe this could do with a bit more discussion in the text as to potential reasons for this noise? In the text it's stated that it is due to the presence of significant knickpoints, but I was confused as to whether basins with knickpoints were excluded or not. It raises the possibility that some of these metrics, especially _H, would be too noisy in real landscapes when additional factors such as variations in lithology, rainfall or uplift are taken into account.

AC: We agree the level of noise observed in Figure 7 is surprisingly high. However, it cannot be explained by knickpoints retreats because we excluded all associated basins from our analysis. Action: Following RC1, we now assess the standard deviation of

cross-divide metric differences along basin perimeter and find that it explains part of the dispersion. We also assess via a "confidence index" the proportion of documented segments when calculating aggressivity metrics. These results are now presented in section 3.

RC2: Lines 275 - 277: I'm also surprised that there is an increase in drainage density with increasing uplift when n = 1 in the model runs. Previous theoretical work by Tucker and Bras [1998] predicted that drainage density should be independent of erosion rate when n = 1, which was then shown by numerical modelling using CHILD by Clubb et al. [2016] (Figures 4 and 5). Furthermore, and more qualitatively, when I have run LEMs with detachment-limited stream power in the past I have generally found that the geometry of the network remains fixed when increasing uplift rate, and only the slopes of the network increase. I think this could be discussed in more detail as to why there is this discrepancy with previous theoretical predictions and numerical modelling results.

AC: Both Tucker and Bras (1998) and Clubb et al. (2016) found that drainage density is independent of erosion rates with n=1 (and therefore, uplift rate in steady-state topography) when using a linear formulation for diffusion. However, when using a formulation that takes into account a threshold slope for hillslope, both studies show an inverse relationship between drainage density and uplift rate. Thus, the results exposed in our study are consistent with these previous theoretical works.

RC2: Line 279: 'we obtain no significant changes in the relationship between the calculated aggressivity metrics and the E/U ratio for uplift rates...' I'm not sure I agree with this statement - from Fig 7, some of the distributions look quite different for U = 0.5 and U = 2 mm/yr, especially for _H. This may be just because of the smaller basins, or that it's difficult to get a sense of how dense the data are in the center of the plot.

AC: We agree the assertion may be misleading. We observe a clear decrease of dispersion for higher uplift rates. However, the trend remains the same regardless of the uplift rate value. In any event, results must be affected by a threshold area Ac fixed
to a non-zero value. Action: We present detailed results concerning sensibility to uplift rate using a Ac = zero in section 4.1.1.

RC2: Lines 276 and 287: I think it would strengthen the paper to quantify this change in 'river channelization' proposed by the authors. At the moment this appears to be a qualitative statement which is difficult to verify from the current figures and analysis. It's an interesting result that changing the value of Sc influences drainage density, which makes sense from the theory and has implications for real landscapes where Sc is difficult to determine. I think more could be made of this, and suggest simply calculating drainage density for the different model runs. This would put some more weight behind the statement that increasing drainage density is the mechanism by which changing U and Sc impact the aggressivity metrics.

AC: We agree. However, the updated model using Ac = 0 does not display significant discrepancies when Sc varies. As pointed out by RC1, Ac > 0 can artificially control hillslope length and drainage density. Letting U vary does have an effect on drainage density, though, and we agree this should be explored. However, we think that this topic may deserve a dedicated study that is beyond the scope of the present work.

RC2: Lines 314 - 315: I don't understand what are the 'expected quadrants' that the basins are being compared to here. Is this compared to the reference model, or compared to Willett et al. (2014)?

AC: We agree we did not state our terminology clearly. We meant 'expected when comparing to the reference model'. Action: We clarified this by adding the following paragraph: "In agreement with cross-divide metrics tested by Forte and Whipple (2018), graphs in Figure 7 must be divided into four quadrants. Aggressor (victim) basins have negative (positive) $\Delta\chi\_av$ and $\Delta H\_av$ values and conversely positive (negative) $\Delta G\_av$ value (Fig. 2). Theoretically, aggressor (victim) basins have higher (lower) denudation rates than the underlying uplift rate." We also explicitly labeled the aggressors and victims quadrants in the associated figures.

**ESurfD**

---

## Author Comment (AC3) · 4 Oct 2019

C1: I enjoyed reading this manuscript and appreciate the strong effort at quantifying intrabasin variability under regional quasi-steady state conditions. It's an important problem and I think this is a very good approach with interesting results. I just wanted to draw the authors' attention to our paper from last year which they might find relevant. We quantified divide migration from event-triggered landslides and found that, although divides generally moved in directions predicted by cross-divide gradients in the Gilbert metrics (which we have attempted to place in the context of progress toward regional steady-state in Taiwan), landslide stochasticity introduces a lot of complications that

would be especially pronounced in small basins. Additionally, we had similar struggles with the use of delta Chi, finding that it works well at predicting divide migration where identifying base level is very simple and poorly where it is not. I thought this was relevant to mention, as many of our conclusions drawn from cross-divide observations at the timescale of a single earthquake/storm agree with the authors' long timescale basin-scale observations. Here is the paper in question: Dahlquist, M. P., West, A. J., and Li, G., 2018, Landslide-driven drainage divide migration: Geology, v. 46, no. 5, p. 403–406, doi:10.1130/G39916.1.

AC : Thank you for your interest. We agree and add this reference in the introduction part of our revised manuscript.

---

## Author Response (AR1)

[revised manuscript text omitted]

455    For victim basins ($\Delta G < 0$), $c_1 \sim 0.05$ and $c_2 \sim -0.5$, whereas for aggressor basins $c_1 \sim -0.14$ and $c_2 \sim 1$. This provides a new additional guideline for the design of sampling strategies in terms of basin size. For instance, considering a quasi-steady state mountain belt with an uplift rate of 1 mm.yr$^{-1}$, the minimum basin area required for an erosion rate variability

lower than 0.5 mm.yr⁻¹ is ca. 1 km² for victim basins and 30-40 km² for aggressor basins.with $c_1$ and $c_2$ two parameters that depend on balance between erosion processes, uplift rate and state of evolution of the landscape.

**5 Conclusions**

Calculations from a Landscape Evolution Model assuming spatially uniform uplift, rock strength and rainfall confirm that the concept of topographic steady state is relevant at the scale of entire mountain belts, but represents an oversimplification at the scale of individual watersheds. Our simulations underline the role of divide mobility on deviations from equilibrium, which can lead to significant differences between tectonic uplift rate and basin-wide denudation rates even if an overall topographic steady state is achieved at large scale.

To better assess these deviations, we propose new basin-averaged aggressivity metrics $-\Delta\chi_{av}$, $\Delta G_{av}$ and $\Delta H_{av}$ - based on the approach of Willett et al. (2014) and Forte and Whipple (2018). They include mean cross-divide contrasts in channel heads $\chi$, local slopegradient and elevation at the scale of entire river basins.height. From our calculations, contrasts in channel head elevation appear$\Delta\chi_{av}$ is the most reliable aggressivity metric to be weakly sensitive to assess local disequilibrium, whereas, but is highly depend on the basin denudation to 
[revised manuscript text omitted]

---

## Author Response (AR2)

RC1: I've completed my review of Sassolas-Serrayet et al's paper entitled 'Estimating the disequilibrium in denudation rates due to divide migration at the scale of river basins. 'In this manuscript, the authors analyze a series of landscape evolution models to try to ultimately assess the ability for one to measure the background rock uplift rate through basin averaged erosion rates in the presence of divide migration and find that there is a drainage area dependence on the ability to do so. Using this, they are able to provide some recommendations for planning of sampling for things like cosmogenic erosion rates. This study seems timely and interesting and is a good fit for Earth

Surface Dynamics. While I think the general results are strong and consistent with what others have shown (or rather what is implied by the results others have shown), I do have some concerns about whether they've biased their detailed results with the specifics of their model setups. Much hinges on whether or not the models were ran with an imposed threshold area or not (it is a little ambiguous in the text, so it's possible that my concerns are for naught and I simply misunderstood what they meant). These concerns (along with other comments that the authors hopefully find helpful) are outlined in my detailed comments below.

AC: Thank you for your constructive comments. The remarks concerning the potential bias associated with the use of a threshold area Ac greater than zero leads to a great number of modifications, especially in the result and discussion sections. However, the results we obtained by using a Ac value equal to zero does not affect the main finding of our study: divide mobility can lead to significant differences between tectonic uplift and basin-wide denudation rates even if topographic steady state is achieved at large scale. We updated the figures in order to represent the new results and take into account the referee's comments. We hope this offers an improved and clearer version of our manuscript.

RC1: L37-38: Probably important to clarify that the divides are migrating in response to the same change (it's implied, but not explicitly stated in the way you word it).

AC: Thank you for this remark. Indeed, the persistence of migrations is not related with change of other factors (i.e. tectonic or climate) during the landscape evolution. Action: We modify the concerned paragraph as following: "Although rivers exhibit a rapid adjustment to tectonic or climatic changes to maintain their profiles, Whipple et al. (2017) show that divides continue to migrate over time periods of 106-107 years as response to the same changes. This suggests that long-term transience might be pervasive in the planar structure of landscapes, even in the absence of new variations in landscape characteristics or forcings (e.g. tectonic or climate) (Hasbargen and Paola, 2000; Hasbargen and Paola, 2003; Pelletier, 2004)."

**ESurfD**
RC1: Equation 1: I think the top expression (U +  $(dz/dt)_fluv$ ) should be for A > A sub c, correct?

AC: Thank you for pointing out this inconsistency. We use a value of Ac equal to 0 in the revised version of the manuscript in order not to separate hillslope and fluvial domains. That way diffusion and fluvial erosion affect simultaneously every pixel in the model. Action: we removed the condition that regards Ac in the equation 1.

RC1: L119 – 121: Is the stated critical drainage area applied in the model (i.e. the model is run with the rule set such that diffusion is only applied where A<A sub c and incision is only applied where A>A sub c) or is this simply the threshold area used for extracting the channels (and thus the channel heads) for analysis? My (anecdotal) experience has been that running TTLEM (or really any model that allows you to do so) with an explicit critical drainage area that is built in where it defines where incision/diffusion is applied can produce some odd behaviors and very odd drainage networks. If you were running TTLEM with the critical drainage area option turned on. did you experiment with the sensitivity of your results to turning this off (i.e. setting it to 0)? It's important to note that this option is kind of atypical, i.e. it is not something allowed in CHILD, LandLab, etc. I think more importantly, if you are running with Ac (or AreaThresh as it's named in the TTLEM setup) set to a value greater than zero you are artificially controlling the length of the hillslope (it would otherwise be set by the combination of K and D values you provide) and thus controlling the length scale over which the landscape responds to the divide migration (as this is happening mostly in the hillslopes based on prior results). This may in turn reflect some of your other results (e.g. erosion rates as a function of drainage area, etc).

AC: We agree. We tested the sensivity of our results with Ac equal to 0. Results confirm the referee's intuition, namely that a value of Ac greater than zero controls the length of hillslopes and thus produces odd behaviors. In the revised version of the manuscript, we use Ac = 0 and modified the text and figures accordingly.

**ESurfD**
RC1: L125-126: This is the theoretical time to steady-state following a perturbation and assuming a fixed drainage area, which is not really applicable to the time to steady-state for a model ramping up (i.e. running from an initial random noise, low elevation topography to a stabilized topography). I don't think this necessarily matters that much to your results as you are basically just exploiting the fact that this portion of a model run has a lot of drainage reorganization, but I would be careful about equating these.

AC: We agree. Action: We removed the sentence.

RC1: L205-207: This pattern in erosion rates as a function of drainage area pretty much follows from the observation discussed in Forte & Whipple 2018, namely that if considering simulated landscapes experiencing progressive divide motion (as opposed to discrete captures), the erosion rate contrasts across divides is very spatially limited to areas very near the divides (essentially hillslopes), thus as you move to larger drainage areas, the ability to 'see' this across divide contrast in erosion rate in basin averaged values would be expected to decrease as the signal is diluted by more and more of the drainage basin eroding very near the background erosion rate / uplift rate.

AC: We agree. Action: We added the sentence "As exposed by Forte & Whipple (2018), the erosion rate contrasts across divides is spatially limited to areas very near the divides."

RC1: L232-236: This all seems logical, however it might be important to note that basin averaged statistics like this work best when a basin is either uniformly (or at least consistently) either expanding or contracting. This is probably (usually) the case in homogeneous models like the ones you use here, but in either more heterogeneous models or when applied to real landscapes, there is the danger of a basin appearing to be neutral because it is expanding in one direction and contracting in another (i.e. the metrics counterbalance when averaged over the whole basin). This could be especially noticeable when divide migration is driven by a lateral gradient in uplift rate with respect to the main drainage direction. Thus, I would (maybe somewhat self importantly) argue

**ESurfD**
that along-divide metrics are still quite useful to consider.

AC: We agree. Even in homogeneous models like the ones we use, basins may experience both expansion and contraction on different parts of their perimeter and a single metrics may not reflect compound behaviors. Action: We added an analysis of the standard deviation of cross-divide differences in metrics along basin perimeter (Figure 7) and associated details in the main text ("Figure 7c shows that the dispersion is related to the standard deviation of aggressivity metrics,  $\Delta\chi$ \_std,  $\Delta$ G\_std and  $\Delta$ H\_std. In other words, basins where different divide segments migrate at different rates or in different directions are more scattered.").

RC1: L249: This is also consistent with what Forte & Whipple 2018 saw in natural landscapes, i.e. across divide contrasts in elevation were usually equivocal in terms of indicating potential for drainage divide motion compared to the other metrics.

AC: We agree. Our updated model shows the metric based on local slope seems less sensitive to basin-wide denudation from divide migration for area with rock uplift rates ( $\leq 0.1 \text{ mm/yr}$ ). Conversely, even if it depends on the average elevation at regional equilibrium stage, the metric based on elevation seems more relevant. However, this metric would likely display significant noise in natural landscapes.

RC1: L254-255: Why do any of these basins have knickpoints? You're applying a constant uplift rate and progressive divide motion shouldn't really impart knickpoints onto any profiles. Are they coming from captures? While I understand the logic of ignoring basins with knickpoints, it's more that I wonder if the presence of knickpoints in this is suggesting that there may be some stability issues. One of the behaviors of the TVDFVM algorithm is to keep and accentuate any knickpoint (even if those are developed through numerical instability), so it would be good to try to diagnose why there are knickpoints in the first place to rule out model instability. You could try running one of your models with the same exact setup but using the implicit (fastscape) algorithm that's built into TTLEM and see if you also are getting knickpoints.

**ESurfD**
AC: We agree, for uniform models knickpoints appear counter-intuitive. However, we observed two processes that produced actual knickpoints (1) the uplift of the initial plateau leads to the propagation of major knickpoints controlled by the edge of the plateau until  $\sim$ 2Myr and (2) as you suggest, discrete stream captures occur occasionally. Overall, the great majority of basins (and therefore drainages) are not affected by any knickpoints. Action: We clarified this point by adding the sentence "In our simulations, knickpoints may develop due to (1) the dissection of the initial flat surface or (2) discrete drainage captures (see Sec. 3.1).".

RC1: L274-278: I'm a bit confused by this statement. From my own simulations with TTLEM, if you're starting with the same random noise and keeping everything else the same (i.e. not changing the length of timestep, etc), the drainage network evolution will be pretty much the same regardless of uplift rate. I would expect changes in the effective drainage density to manifest more in response to changes in the diffusion constant or ratios between K and D.

AC: We agree. This behavior was driven by Ac > 0 and our updated model shows a different pattern. We now observe no differences in general drainage geometry. However, we observe an inverse relationship between uplift rate and drainage density consistent with previous theoretical works. Tucker and Bras (1998) found a similar relationship when using a formulation of diffusive processes that includes a threshold slope Sc, as we use in our simulation.

RC1: L283-285: Again, this might be a result of setting the area threshold to a non zero value (assuming you did). I'm also not sure how to interpret Sc if the area threshold is set to a non zero value, because you're artificially controlling the hillslope length and thus (I think) artificially controlling the maximum slope that can develop anyways. If I misunderstood your discussion of A sub c earlier and you were not running with A sub c set to a non zero value, feel free to ignore this comment.

AC: We agree, this is indeed driven by Ac>0. Action: We modified this part of the

**ESurfD**
discussion based on the updated results.

RC1: L340-345: This is neat, though (and as you mention in the following sentence) it would be interesting to think about the applicability of this in more heterogeneous environments where, as I mentioned before, there's a greater chance that the basin averaged metric could be misleading (i.e. a low average metric because of coexisting drainage area loss and gain on different sides). Maybe a valuable approach to consider would be including the standard deviation (or min and max) in the basin averaged metric to try to capture this potential variability without having to look at the divide segments in detail?

AC: We agree. As we explained in the previous comment concerning L232-236, we assess the variability of cross-divide contrast in metrics by calculating the standard deviation of metrics along basin perimeter. This allows to discriminate between basins that are homogeneously stable and those that display both expansion and contraction along their perimeter. Action: See RC1: L232-236.

RC1: L349-372: This is a good thing to focus on, but I think you could add an interesting discussion here of considering the sampling strategy with regards to the goal of the study. At present, this gives a (valuable) set of ideas for how large a basin needs to be to get an accurate assessment of the uplift rate from the erosion rate, but alternatively you could point out that this gives you a sense of the size of basin you need to target if you're explicitly interested in measuring divide migration rates, i.e. larger basins are not going to be helpful. Similarly, this speaks to the need to pre assess basins for their potential divide mobility before sampling (if your intention is to get at background uplift rate), i.e. if either the divide segment or basin average metrics suggest no divide mobility, you don't need to worry about the size of the basins as much (except for all the other concerns we already have, that you mention).

AC: We thank you for this pertinent suggestion. Action: We developed this aspect as a new full paragraph in the discussion section.

**ESurfD**
Earth Surf. Dynam. Discuss., https://doi.org/10.5194/esurf-2019-31-AC2, 2019 © Author(s) 2019. This work is distributed under the Creative Commons Attribution 4.0 License.

ESurfD
RC2: The paper in review presents a series of numerical models set up to investigate the impact of drainage divide migration on landscape denudation rates, as well as both presenting new topographic metrics and testing the ability of previous ones to detect divide migration. The authors then apply these metrics to provide some useful constraints for basin size for CRN sampling. I think the paper is interesting, well written, and builds well on previous work that quantifies the extent of drainage divide migration across landscapes, given the many papers that have been published in the last few years on the topic. However I have some concerns about the setup of the model, in

particular the use of a parametrised critical area threshold. I therefore think it will be very suitable for publication in ESURF provided that the comments detailed below can be addressed.

AC: Thank you for your positive and constructive comments. We carefully addressed every remark you formulated. Your concerns about threshold area Ac to distinguish hillslope and fluvial domains were justified. In the revised manuscript we produce a new set of results using simulations with no threshold area (Ac=0). This change largely corrects the atypical behaviors underline by both reviewers concerning the length of hillslope, the dependency of drainage network evolution to rock uplift, and the meaning of Sc, but it does not modify the major results concerning the impact of drainage migration on basin-wide denudation rates. We provide a revised version of the manuscript where we present our new results and addressing the referees comments.

RC2: Line 34 - 35: repetition of 'modern landscapes' twice in the sentence.

AC: We agree. Action: We reworded the sentence accordingly.

RC2: Eq 1: I think the fluvial expression should be A > Ac?

AC: We agree. Action: As modified our model with Ac = 0, we removed this condition in Equation 1. See RC1 for details.

RC2: Eq 1: Similar to Reviewer 1, I agree that the model setup described in equation 1 makes it seem like the position of channel heads and resulting drainage divide metrics will be determined by the critical area threshold (Ac) parameter. I'm not convinced that, in real landscapes, there is a critical area threshold for determining channel head location or, if there is, that this should be fixed across the landscape as a whole. This setup seems a bit unintuitive to me - why not just combine fluvial incision with hillslope diffusion at each node and eliminate the need for an Ac parameter in the LEM at all?

AC: We agree. See RC1's comment. Action: See RC1.

RC2: Line 118: Much work has shown that in many real landscapes it is most likely that
n > 1 [e.g. Lague, 2014; Harel et al., 2016]. This will again have a significant impact on the distribution of slopes and erosion rates within the model landscapes, and may influence the calculation of the divide migration metrics. I think it would be useful to run some test landscapes where n > 1 to determine i) the impact this has on the variability of erosion rate with basin area; and ii) the impact on the aggressivity metrics.

AC: We agree this would be a very interesting alley to explore. However, this would constitute an entirely new study that would go far beyond the scope of the present study. Action: We mentioned this as a perspective in the conclusions.

RC2: Line 121: Following on from this, I would suggest running a sensitivity analysis changing Ac in the model setup to see what effect this has on the calculation of divide migration metrics.

AC: We agree this could be a worthy path to follow. However, following comments from both referees we opted to set Ac = 0 in line with the majority of works published previously in order to produce results that can be compared.

RC2: Line 125: How is it determined whether the model runs have indeed reached steady state?

AC: As expected, the model does not actually reach a perfect steady state but a quasistatic equilibrium. First, we study the reorganization of the drainage after an initial perturbation keeping in mind that the drainage network shall not be affected by the initial model geometry. Next, in our simulations, we consider that the landscape reaches a regional steady state once transient topographies are completely eroded, i.e. once the mean model elevation remains constant and when the average denudation rate and the imposed rock uplift rates deviate by less than 2%. Finally, transient events such as discrete stream captures susceptible to form knickpoints are carefully detected and discarded from the results.

RC2: Line 135: It's a bit unclear how the elevation is calculated. Is this the average

**ESurfD**
elevation for all pixels in the basin at each time step?

AC: Basin-wide denudation rate is calculated using the average of differences of elevation in the basin borders during a 10 kyr interval.

RC2: Line 162: It would be good to include some more details here of how this averaging is carried out. Is local slope calculated using a moving window (including hillslopes), or is this just the slope of the first order channel? It wasn't clear to me whether the elevation at each channel head was averaged, or whether the first order channel downstream of the channel head was included. Fig S2: I think it would be useful to edit this figure and move it to the main paper, as it was difficult to follow how the aggressivity metric is calculated from the text. I realise some of this is based on previous work, but the averaging of the cross-divide metrics to produce a new metric for each basin is novel in this paper. I didn't understand how the segmenting shown in Fig S2 was done, or how the segment length over which the averaging should be performed is determined.

AC: We agree we did not provide enough detail on this. All metrics are measured at a reference drainage area. Across-divide metric differences are only calculated along divide segments shared by two reference basins. Hence, aggressivity metrics are the averaging of these documented segments along the perimeter of the sampled basin. Action: We reshaped section 2.3.2 and transferred Fig. S2 from the Supp. Mat. to the main text (now Fig. 2).

RC2: Line 221: As well as drainage divide migration, variability in erosion rate between basins could simply result from the transient propagation of knickpoints, especially in the earlier model runs. This seems to be supported by the fact that the variability decreases significantly through time as shown in Figures 4 (a) - (c). 1 What is the evidence that this variability is in fact due to divide migration and not due to transient knickpoint propagation? In the text it's stated that this is shown from Fig 2(e), but I didn't understand how this figure shows that.

**ESurfD**
AC: It is true that transient propagation of knickpoints can affect basin-wide denudation rates. To avoid any erosion signal associated with these transient features, we do not take into account the first stage of the model (

cross-divide metric differences along basin perimeter and find that it explains part of the dispersion. We also assess via a "confidence index" the proportion of documented segments when calculating aggressivity metrics. These results are now presented in section 3.

RC2: Lines 275 - 277: I'm also surprised that there is an increase in drainage density with increasing uplift when n = 1 in the model runs. Previous theoretical work by Tucker and Bras [1998] predicted that drainage density should be independent of erosion rate when n = 1, which was then shown by numerical modelling using CHILD by Clubb et al. [2016] (Figures 4 and 5). Furthermore, and more qualitatively, when I have run LEMs with detachment-limited stream power in the past I have generally found that the geometry of the network remains fixed when increasing uplift rate, and only the slopes of the network increase. I think this could be discussed in more detail as to why there is this discrepancy with previous theoretical predictions and numerical modelling results.

AC: Both Tucker and Bras (1998) and Clubb et al. (2016) found that drainage density is independent of erosion rates with n=1 (and therefore, uplift rate in steady-state topography) when using a linear formulation for diffusion. However, when using a formulation that takes into account a threshold slope for hillslope, both studies show an inverse relationship between drainage density and uplift rate. Thus, the results exposed in our study are consistent with these previous theoretical works.

RC2: Line 279: 'we obtain no significant changes in the relationship between the calculated aggressivity metrics and the E/U ratio for uplift rates...' I'm not sure I agree with this statement - from Fig 7, some of the distributions look quite different for U = 0.5 and U = 2 mm/yr, especially for \_H. This may be just because of the smaller basins, or that it's difficult to get a sense of how dense the data are in the center of the plot.

AC: We agree the assertion may be misleading. We observe a clear decrease of dispersion for higher uplift rates. However, the trend remains the same regardless of the uplift rate value. In any event, results must be affected by a threshold area Ac fixed

**ESurfD**
to a non-zero value. Action: We present detailed results concerning sensibility to uplift rate using a Ac = zero in section 4.1.1.

RC2: Lines 276 and 287: I think it would strengthen the paper to quantify this change in 'river channelization' proposed by the authors. At the moment this appears to be a qualitative statement which is difficult to verify from the current figures and analysis. It's an interesting result that changing the value of Sc influences drainage density, which makes sense from the theory and has implications for real landscapes where Sc is difficult to determine. I think more could be made of this, and suggest simply calculating drainage density for the different model runs. This would put some more weight behind the statement that increasing drainage density is the mechanism by which changing U and Sc impact the aggressivity metrics.

AC: We agree. However, the updated model using Ac = 0 does not display significant discrepancies when Sc varies. As pointed out by RC1, Ac > 0 can artificially control hillslope length and drainage density. Letting U vary does have an effect on drainage density, though, and we agree this should be explored. However, we think that this topic may deserve a dedicated study that is beyond the scope of the present work.

RC2: Lines 314 - 315: I don't understand what are the 'expected quadrants' that the basins are being compared to here. Is this compared to the reference model, or compared to Willett et al. (2014)?

AC: We agree we did not state our terminology clearly. We meant 'expected when comparing to the reference model'. Action: We clarified this by adding the following paragraph: "In agreement with cross-divide metrics tested by Forte and Whipple (2018), graphs in Figure 7 must be divided into four quadrants. Aggressor (victim) basins have negative (positive)  $\Delta \chi_a v$  and  $\Delta H_a v$  values and conversely positive (negative)  $\Delta G_a v$  value (Fig. 2). Theoretically, aggressor (victim) basins have higher (lower) denudation rates than the underlying uplift rate." We also explicitly labeled the aggressors and victims quadrants in the associated figures.

**ESurfD**
|----|---------------------------------------------------------------------------------------------------------------------------------------------------------------------------------------------------------------------------------|---|-------------------------------------|
|    | Estimating the disequilibrium in depudation rates due to divide                                                                                                                                                                 |   |                                     |
|    | migration at the goals of river basing                                                                                                                                                                                          |   |                                     |
|    | niigrauon at the scale of river basins.                                                                                                                                                                                         |   |                                     |
|    | Timothée Sassolas-Serrayet 1 , Rodolphe Cattin 1 , Matthieu Ferry 1 , Vincent Godard 2 , Martine Simoes 3                                                                |   | (Mis enforme: Français(France)      |
|    | 1 Géosciences Montpellier, Université de Montpellier <del>and CNRS UMR 5243, Montpellier 34095, France.</del>                                                                                          |   | Mis en for me: Français(France)     |

[revised manuscript text omitted]

**Mis enforme: Anglais(ÉtatsUnis)**

**Mis enforme: Anglais(ÉtatsUnis)**

|   | Mis enforme: Couleurde police :
Noir |
|---|-----------------------------------------|
|   |                                         |
| { | Mis en for me: Couleurde police :       |

Noir

heads (Clubb et al., 2016). Hence, we use a constant value of  $A_{ref}$  set to 1 km2. The parameter  $\chi$ , is an integral function of position along the channel network (Perron and Royden, 2012) described by the equation:

180
$$\chi = \int_{x_b}^x \left(\frac{A_0}{A(x)}\right)^{\frac{m}{n}} dx$$

185

(4)

where A(x) is the upstream drainage area at the location x,  $A_0$  is an arbitrary scaling area set to 1 km2. The *m* over *n* ratio refers here to the reference concavity of an equilibrated river profile. Its value is set to 0.5 in accordance with the model parameters. For each independent drainage network, we integrate  $\chi$  from the outlet  $x_{b_1}$  located at the model boundary (< 1 m high), to the channel heads. We then Local gradient is determined for each DEM pixel from its eight-connected neighbors.

Height is simply extracted from the DEM.

Then, we calculate the difference in channel head  $\chi$ , local slope of metrics  $(\Delta \chi, \Delta G$  and elevation  $\Delta H$ ) across each first order basin the segments of divide segment. The shared by two reference basins. Finally, the aggressivity metric is finally obtained by averaging these first order across-divide differences along the perimeter of each extracted sampled basin (Fig. S-2). This way, the sign of the aggressivity metric in a basin corresponds to the difference of the averaged value of considered metric

- 190 way, the sign of the aggressivity metric in a basin corresponds to the difference of the averaged value of considered metric (Channel head  $\chi$ , slope or elevation difference ( $\Delta \chi \Delta G$  and  $\Delta H$ ) in this basin with respect to hisits neighbours. This method has the advantage to ponderate the weight of individual divide segments by the number of pixels they contain, and then to provide a robust assessment of the basin aggressivity. Aggressivity metrics based on  $\chi$ , G and H are hereafter referred to as  $\Delta \chi_{av} \Delta G_{av}$  and  $\Delta H_{av}$ , respectively, which provide an accurate assessment of the basin aggressivity. However, due to
- 195 topology issues, some parts of the perimeter of the sampled basins may be not shared by two reference basins (Fig. 2). We quantify this incompleteness by assessing the ratio of documented pixels over the total amount of pixel along the basin perimeter. We refer to this ratio as the "confidence index" CI, assuming that an higher CI is associated with a more robust basin aggresivity assessment.

**Mis enforme: Anglais(ÉtatsUnis) Mis en forme: Anglais(ÉtatsUnis) Mis enforme: Anglais(ÉtatsUnis)**

**3 Results**

**200 **3.1 Evolution of reference model**

A detailed analysis of the DEM suggests that during the initial phase, the flat initial surface (Fig. 2a3a) is progressively uplifted to form a plateau. At the same time the edges of this plateau are gradually regressively eroded by drainage networks that spread from the base level toward the center of the model (Figs. 2b3b and c). This transient landscape is completely dissected after 2 Myr. From this time and until the end of the simulation, landscape changes are mainly due to competition

205 between watersheds, resulting in continuous divide migrations with decreasing intensity as the model is moving toward a total topographic equilibrium (Fig. 2d3d to f; Supplementary Video n°1).

WeTo define the time period of regional steady state, we measure the average elevation, the maximum elevation and the average denudation rate over the entire model for each time step (Fig. 3a) and 4a). We identify two distinct stages during the evolution of our reference simulation. During the first million years, due to long wavelength topographic building, the calculated landscapes are far from steady state. This leads to a major increase of the mean elevation from 0 m to ca. 70075

calculated landscapes are far from steady state. This leads to a major increase of the mean elevation from 0 m to ca. 70075
 m. In a second stage, this trend reverses and the mean elevation decreases asymptotically toward ca. 60060 m until the end of the simulation.

The evolution of the maximum elevation follows the same pattern but can be affected by temporal changes in the location and altitude of higherhighest peaks. The maximum elevation increases between 0 and ca.  $\frac{2200250}{2200250}$  m over the first  $\frac{2}{2}$  Myr

215 (Figs. 3a3Myr (Fig. 4a) then decreases progressively to reachremain at ca. 1600200 m atduring the endrest of the simulation. We compute the average denudation rate from the tectonic prock uplift rate and from average elevation change over the entire model between two time steps ÷:

$$(\Delta z / \Delta t)_{av} = U - E_{av} ,$$

220

where  $(\Delta z/\Delta t)_{av}$  is the average surface uplift over the entire model on a time-step  $\Delta t$ , U-is the imposed uniform uplift rate Mis enforme: Anglais(ÉtatsUnis) Mis enforme: Anglais(ÉtatsUnis) (0.1 mm.yr-1) and  $E_{av}$  is the average "real" denudation rate. During the first 0.25 Myr, the mean denudation rate falls Mis enforme: Anglais(ÉtatsUnis) abruptly from ca. 0.6 mm, yr-1 to nearly 0 mm, yr-1 as a consequence of diffusion over the initial flat topography. After that Mis enforme: Anglais(ÉtatsUnis) time and until the first 1 Myr, the mean denudation rate increases but remains lower than the uplift rate, leading to the 225 increase in average elevation over this time period. Next in In the following 0.51 Myr. Fax exceeds the uplift rate to reach up Mis enforme: Anglais(ÉtatsUnis) to 1.030.104 mm.yr-1. It then before it gently decreases to 0.1 mm.yr-1 duringuntil the restend of the simulation. This shows that topography tends to - but never reaches - a strict steady state over the simulation time. Abrupt changes in  $E_{av}$  after ca. 2Mis enforme: Anglais(ÉtatsUnis) Myr, 2.5 Myr and 3.2 Myr can also be highlighted (Fig. 3b). These brief variations 2.5, 3.5, 4.5 and 9.5 Myr (red circles in Figure 4b) are related to major local captures in the drainage network, which can be observed during the model evolution 230 (red circles in Fig. 2d3e and f and Supplementary Video n°1). On the basis of Based on these results, we will consider that quasi-a regional topographic steady-state is reached between 1.5 and 2 Ma, when the plateau relict topography is totally eroded and  $F_{av}$  begins to decrease (Figs.  $\frac{23}{24}$  and  $\frac{34}{24}$ ). This time is Mis enforme: Anglais(ÉtatsUnis) consistent with the time required to reach topographic steady state proposed from models with constant uplift rate and no horizontal advection (Willett et al., 2001). 3.2 Basin-wide denudation rates variability 235 We calculate basin-wide denudation rates E upstream of each stable drainage network confluence after 2.5 Myr, 5 Myr and Mis enforme: Anglais(ÉtatsUnis) 10 Myr of simulation (Figs. 4a, b and c). As explained in the method section, we compiled the results obtained for five runs ampled basins.5a, b and c, in order to

(5)

| I   | respectively) Regardless of the duration, we observe a significant variability in the calculated depudation rates depending on                                                            |
Mis on former Anglais/Poyaume-Un  |
|-----|-------------------------------------------------------------------------------------------------------------------------------------------------------------------------------------------|---------------------------------------|
| 240 | basin size. This As exposed by Forte & Whipple (2018), the erosion rate contrasts across divides is spatially limited to areas                                                            |                                       |
|     | very near the divides. Thus, the variability is maximum for small basins (ca. 1 km 2 ) and decreases with increasing basin area                                                |
Mis on former Anglais/Poyaume-Uni |
|     | In our approach small basins are nested in larger ones. Hence these results can be related to the averaging of depudation                                                                 |                                       |
|     | rates along the drainage network in agreement with the measurements of Matmon et al. (2003b). This variability along                                                                      |                                       |
|     | decreases with time (Eise 4a h and 5a c). For basing with an average of depudding relative to the writing rote II the E/II ratio                                                          |                                       |
| 245 | decreases with time (Figs. $\frac{4}{40}$ , $\frac{1}{100}$ and $\frac{3}{24}$ -c). For basins with an excess of denudation relative to the uplit rate $U$ , the $E/U$ ratio              | Mis enforme: Anglais(EtatsUnis)       |
| 245 | can reach up to $\frac{32.5}{2.5}$ after 2.5 Myr but only 2 after 5 Myr and $1.57$ after 10 Myr. Basins with a denudation excess that stand                                        | Mis en forme: Anglais(ÉtatsUlhis)     |
|     | out of the general trend at 10 Ma (Fig. 5c) are associated with a capture event visible in Figure 4b. For basins with a deficit of                                                        |                                       |
|     | denudation, this the evolution of the ratio is less obvious. It can be lower than 0.255 after 2.5 Myr, but increases slightly to                                                          |                                       |
|     | 0.4 after 56 until 10 Myr. These results reflect a significant spatial variability of the difference between basin-wide                                                                   |                                       |
|     | denudation rates and 0.5 after 10 Myruplift rate. To assess more accurately the temporal evolution of this variability, we                                                                |                                       |
| 250 | calculate E every 0.5 Myr for three distinct categories of basin sizes: 1-2 km 2 , 10-20 km 2 and 100-200 km 2 . We then                          |
|     | estimate the mean absolute deviation (MAD) from the uplift rate by considering separately basins with a denudation in                                                                     |                                       |
|     | excess or in deficit (Fig. 445d). Until 1.5 Ma, basins are located on the plateau where denudation rate is null. This leads to a                                                          |                                       |
| 1   | low MAD for basins with a denudation deficit and to the absence of basins with a denudation excess. After 1.5 Ma, basins in                                                               |                                       |
|     | deficit exhibit an $\frac{\text{asymptotic}}{\text{increase}}$ in MAD from nearly $-0.215$ to $-0.0504$ mm.yr -1 , regardless of the area class considered.                    |                                       |
| 255 | For basins in excess, the MAD value decreases through time, depending on drainage area : from ca. $0.\frac{28 \text{ mm.yr}^4 25}{25}$ to ca.                                             |                                       |
|     | 0.0507 mm.yr -1 for basins with an area of 1-2 km 2 and 10 20 km 2 ; from ca. $0.142$ mm.yr -1 to ca. $0.0507$ mm.yr -1 for basins |                                       |
|     | with an area of 100-200km?. These results reflect a significant spatial variability of 10-20 km 2 and from ca. 0.7 to ca. 0.04                                                 |                                       |
|     | mm.yr -1 for the difference between basin-wide denudation rates and uplift rate in our reference models-largest basins. We                                                     |                                       |
|     | also see a coherent evolution of this difference over the simulation time, consistent with the model progression toward a total                                                           |                                       |
| 260 | topographic equilibrium.                                                                                                                                                                  |                                       |
|     | The spatial variability of the denudation rates is neither homogeneous nor randomly distributed (Fig. 5a6a). The location of                                                              |                                       |
|     | drainage basins with denudation rates far from the equilibrium value of $0.1$ mm.yr -1 coincides with migrating drainage                                                       |                                       |
|     | divides (Fig. 2e3d) and with cross-divide contrasts in headwater channel $\chi$ , slope gradient and elevation height (Figs. 5b, e and                                                    |
Mis enforme: Anglais(ÉtatsUnis)   |
|     | 6b-d). Following Willett et al. (2014) and Forte and Whipple (2018), the divide migrations predicted by these contrasts are                                                               |                                       |
| 265 | consistent with the direction of divide mobility obtained from our model. One may note that the higher the contrast in these                                                              |                                       |
| 200 | parameters across the divide the higher the deviation of the depudation rate from the unlift rate and therefore from                                                                      |                                       |
| I   | tonographic equilibrium These None of sampled basin in this data-set contains a knickpoint. Thus, these results based on                                                                  |                                       |
|     | simulations assuming uniform and constant properties as well as constant boundary conditions confirm that the dispersion                                                                  |                                       |
|     | observed in depudation rates is primerily controlled by divide migration. Basing that around (christ) show higher (lower)                                                                 |                                       |
| 270 | donudation rates compared to unlift rate and an homefter referred to accompare (victime) following the territed or                                                                        |                                       |
| 270 | denudation rates compared to upint rate, and are nerearter referred to as aggressors (victims), following the terminology                                                                 | ()                                    |
|     | adopted by whilet et al. (2014).                                                                                                                                                          |

| Willett et al. (2014) showed that the basin-averaged cross-divide contrast in $\chi$ , could be used to deduce an aggressivity
metric for basins. We extend this basin-scale approach to the Gilbert's metrics recently proposed by Forte and Whipple
(2018) including cross-divide contrast in headwater slope and elevation. Hereinafter, we refer to these aggressivity metrics
based on cross-divide contrast in headwater $\chi$ , headwater slope (gndient) and headwater clevation (height) as $4\chi$ - $4G$ and
$4H_{\rm gradient}$ and elevation, mespectively.Mis enforme: Anglais(États205(2018) including cross-divide contrast in headwater $\chi$ , headwater slope (gndient) and headwater clevation (height) as $4\chi$ - $4G$ and
$4H_{\rm gradient}$ and elevation, mespectively.Mis enforme: Anglais(États206reference models after a simulation duration of 2.5 Myr. Denudation rates may be affected by knickpoints, which are a
source of transient perturbation at the scale of the catchment. Therefore, in order to focus only on perturbations associated
with drainage divide dynamics, basins that contain knickpoints are ignored. In agreement with cross-divide metrics tested by
Forte and Whipple (2018), aggressorgraphs in figure 7 must be divided into four quadrants. Aggressor (victim) basins have
negative (positive) $4\chi d\chi_{av}$ and $4H dH_{av}$ values and conversely positive (negative) $4G dG_{av}$ value: (Fig. 2). Theoretically,
aggressor (victim) basins have higher (lower) denudation rates than the underlying uplift rate. Therefore graphs in figure 6
must be divided into four quadrants, with aggressors situated in the lower left one, and victims in the higher right one. This
result is verified for ca. 91, 8881 %, 52 %, and 8281 % of basins of aggressivity metric based on headwater $\chi$ , slope and
elevation value: $A_{drav} and AH_{arx} respectively (Figs. 6a, b and c). Sevenal basins depart significantly formthe expected$                                                                                                                                                                                                                                      | Unis)
Unis)
Unis) |
|---------------------------------------------------------------------------------------------------------------------------------------------------------------------------------------------------------------------------------------------------------------------------------------------------------------------------------------------------------------------------------------------------------------------------------------------------------------------------------------------------------------------------------------------------------------------------------------------------------------------------------------------------------------------------------------------------------------------------------------------------------------------------------------------------------------------------------------------------------------------------------------------------------------------------------------------------------------------------------------------------------------------------------------------------------------------------------------------------------------------------------------------------------------------------------------------------------------------------------------------------------------------------------------------------------------------------------------------------------------------------------------------------------------------------------------------------------------------------------------------------------------------------------------------------------------------------------------------------------------------------------------------------------------------------------------------------------------------------------------------------------------------------------------------------------------------------------------------------------------------------------------------------------------------------------------------------------------------------------------------------------------------------------------------------------------------------------------------------------------------------------------------------------------------------------------------------------------------------------------------------------------------------------------------------|-------------------------|
|  <li>metric for basins. We extend this basin-scale approach to the Gilbert's metrics recently proposed by Forte and Whipple</li> <li>(2018) including cross-divide contrast in headwater slope and elevation. Hereinafter, we refer to these aggressivity metrics based on cross divide contrast in headwater x, headwater slope (gradient) and headwater clevation (height) as 4x, 4G and 4H-gradient and elevation, respectively.</li> <li>We here assess the relationship between the E/U ratio and these aggressivity metrics. First, to exclude variability related to both basin area and time, we focus on a single class of basins with a size of 10-202-4 km2 gathered from five computed reference models after a simulation duration of 2.5 Myr. Denudation rates may be affected by knickpoints, which are a source of transient perturbation at the scale of the catchment. Therefore, in order to focus only on perturbations associated with drainage divide dynamics, basins that contain knickpoints are ignored. In agreement with cross-divide metrics tested by Forte and Whipple (2018), negreessorgraphs in figure 7 must be divided into four quadrants, with aggressors situated in the lower left one, and vietures in the higher right one. This result is verified for ca. 91, 8881 %, 52 % and 8281 % of basins for aggressivity metrics denue denue for AG and AH these exhibit significant knickpoints in their durange network that increase measured denutation measure denutation measure denutation for AG and AH. these exhibits significant knickpoints in their durange network that increase measured denutation for the dispersion observed around this first order trend may be explained by a linear relationship (Figs. 6a and b). Part of the dispersion observed around this first order trend may be explained by a metric Anglais(États)</li>                                                                                                                                                                                                                                                                                                                                                | Unis)                   |
| 275 (2018) including cross-divide contrast in headwater slope and elevation. Hereinafter, we refer to these aggressivity metrics
based on cross divide contrast in headwater $g_i$ headwater slope (gradient) and headwater elevation (height) as $d_g$ . $4G$ and
$4H_{ematient}$ and elevation_metricity.
We here assess the relationship between the $E/U$ ratio and these aggressivity metrics. First, to exclude variability related to
both basin area and time, we focus on a single class of basins with a size of $10-202-4$ km 2 gathered from five computed
asource of transient perturbation duration of 2.5 Myr. Denudation rates may be affected by knickpoints, which are a
source of transient perturbation at the scale of the catchment. Therefore, in order to focus only on perturbations associated
with drainage divide dynamics, basins that contain knickpoints are ignored. In agreement with cross-divide metrics tested by
Forte and Whipple (2018), aggressorgraphs in figure 7 must be divided into four quadrants. Aggressor (victim) basins have
negative (positive) $4\chi d_{Xav}$ and $4HAH_{av}$ values and conversely positive (negative) $4GAG_{av}$ value; (Fig. 2), Theoretically,
aggressor (victim) basins have higher (lower) denudation rates than the underlying uplift rate. Therefore graphs in figure 6
must be divided into four quadrants, with aggressors situated in the lower-left one, and victims in the higher right one. This
result is verified for ca. $9H_{*}881$ %, $52$ % and $8281$ % of basins for aggressivity metric based on headwater $\chi$ , slope and
elevation values: $4\chi_{av}\Delta G_{av}$ and $AH_{av}$ , respectively (Figs. 6a, b and e). Several basins depart significantly from the expected
quadrants for $\Delta G$ and $\Delta H$ , these exhibit significant linckpoints in their damage network that increase measured demudation
relationship (Figs. 6a and b). Part of the dispersion observed around this first order trend may be explained by
momention in the calculation of demudation marea and agreensitive metrics $T_{av}$ (D). Command to other metr                                                                               | Unis)
Unis)          |
| based on cross divide contrast in headwater $\chi$ , headwater slope (gradient) and headwater elevation (height) as $\Delta \chi$ , $\Delta G$ and
$\Delta H_{\text{gradient}}$ and elevation, respectively.
We here assess the relationship between the $\underline{F}/\underline{V}$ ratio and these aggressivity metrics. First, to exclude variability related to
both basin area and time, we focus on a single class of basins with a size of $10-202.4$ km 2 gathered from five computed
reference models after a simulation duration of 2.5 Myr. Denudation rates may be affected by knickpoints, which are a
source of transient perturbation at the scale of the catchment. Therefore, in order to focus only on perturbations associated
with drainage divide dynamics, basins that contain knickpoints are ignored. In agreement with cross-divide metrics tested by
Forte and Whipple (2018), aggressorgraphs in figure 7 must be divided into four quadrants. Aggressor (victim) basins have
negative (positive) $4\chi 4\chi_{aw}$ and $4HAH_{aw}$ values and conversely positive (negative) $4GAG_{aw}$ value. (Fig. 2). Theoretically,
aggressor (victim) basins have higher (lower) denudation rates than the underlying uplift rate. Therefore graphs in figure 6
must be divided into four quadrants, with aggresson situated in the lower left one, and victims in the higher right one. This
result is verified for ca. 91, 4881 %, 52 %, and 8281 % of basins for aggressivity metric based on headwater $\chi$ , slope and
elevation values, $\Delta \zeta_{aw} \perp \Delta G_{aw}$ and $\Delta H_{aw}$ , respectively (Figs. 6a, b and e). Several basins depart significantly from the expected
quadrants for $\Delta G$ and $\Delta H_{aw}$ , respectively (Figs. 6a, b and e). Several basins depart significantly from the expected
quadrants for $\Delta G$ and $\Delta H_{aw}$ , respectively (Figs. 6a, b and e). Several basins depart significantly from the expected
quadrants for $\Delta G$ and $\Delta H_{aw}$ , respectively (Figs. 6a, b and e). Several basins depart significantly from the expected
quadrants for $\Delta G$ and $\Delta H_{aw}$ to the dispersion observed around this fi | Unis)
Unis)          |
| 4H-gradient and elevation-respectively.Mis enforme: Anglais(États
Mis enforme: Anglais(États280reference models after a simulation duration of 2.5 Myr. Denudation rates may be affected by knickpoints, which are a
source of transient perturbation at the scale of the catchment. Therefore, in order to focus only on perturbations associated
with drainage divide dynamics, basins that contain knickpoints are ignored. In agreement with cross-divide metrics tested by
Forte and Whipple (2018), eggressorgraphs in figure 7 must be divided into four quadrants. Aggressor (victim) basins have
negative (positive) $4\#Ad_{xw}$ and $4#AH_{wv}$ values and conversely positive (negative) $4GAG_{wv}$ value-r(Fig. 2). Theoretically,
aggressor (victim) basins have higher (lower) denudation rates than the underlying uplift rate. Therefore graphs in figure 6
must be divided into four quadrants, with aggressors situated in the lower left one, and victims in the higher right one. This
result is verified for ca. $91, 8881$ %, $52$ % and $8281$ % of basins for eggressivity metric based on headwater $\chi$ , slope and
elevation values, $\Delta\chi_{wv}$ , $\Delta G_{wv}$ and $AH_{wv}$ , respectively (Figs. 6a, band c). Sevenal basins depart significantly from the expected
quadrants for $4G_{and}$ $4H'$ these exhibit significant knickpoints in their durinage network that increase measured demudation
relationship (Figs. 6a and b). Part of the dispersion observed around this first order trend may be explained by
merovinations in the adjustic first $The chipment of the dispersion observed around this first order trend may be explained bymerovinations in the adjustic first The chipment of the dispersion observed around this first order trend may be explained bymerovinations in the adjustic first The chipment of the order and agreesivity, metrice Fig. Th). Compared to other metrics AH$                                                                                                                                                                                                                                                                                                    | Unis)
Unis)          |
| We here assess the relationship between the $E/U$ ratio and these aggressivity metrics. First, to exclude variability related to
both basin area and time, we focus on a single class of basins with a size of $\frac{10}{10} \cdot 202_{-4}$ km 2 gathered from five computedMis enforme: Anglais(États280reference models after a simulation duration of 2.5 Myr. Denudation rates may be affected by knickpoints, which are a
source of transient perturbation at the scale of the catchment. Therefore, in order to focus only on perturbations associated
with drainage divide dynamics, basins that contain knickpoints are ignored. In agreement with cross-divide metrics tested by
Forte and Whipple (2018), aggressorgraphs in figure 7 must be divided into four quadrants. Aggressor (victim) basins have
negative (positive) $4\chi A_{\chi_{av}}$ and $4HAH_{av}$ values and conversely positive (negative) $4GAG_{av}$ value: (Fig. 2). Theoretically,
aggressor (victim) basins have higher (lower) denudation rates than the underlying uplift rate. Therefore graphs in figure 6
must be divided into four quadrants, with aggressors situated in the lower left one, and victims in the higher right one. This
result is verified for ca. $91, 8881, \%, 52, \%$ and $8281, \%$ of basins for aggressivity metric based on headwater $\chi$ , slope and
elevation values, $\Delta_{\chi_{av}} \Delta G_{av}$ and $\Delta H_{av}$ , respectively (Figs. 6a, b and e). Several basins depart significantly from the expected
quadrants for $\Delta G$ and $\Delta H$ . these exhibit significant knickpoints in their drainage network that increase measured denudation
relationship (Figs. 6a and b). Part of the dispersion observed around this first order trend may be explained by
more explained by
more in the calculation of denudation area and aggressivity metrics Fig. 7b) Command to other metrics $AH$                                                                                                                                                                                                                                                                                                         | Unis)
Unis)          |
| both basin area and time, we focus on a single class of basins with a size of $\frac{10 - 202 - 4}{10 - 202 - 4}$ km 2 gathered from five computed
reference models after a simulation duration of 2.5 Myr. Denudation rates may be affected by knickpoints, which are a
source of transient perturbation at the scale of the catchment. Therefore, in order to focus only on perturbations associated
with drainage divide dynamics, basins that contain knickpoints are ignored. In agreement with cross-divide metrics tested by
Forte and Whipple (2018), aggressorgraphs in figure 7 must be divided into four quadrants. Aggressor (victim) basins have
negative (positive) $4 \pm 4 \lambda q_{xav}$ and $4 \# A H_{av}$ values and conversely positive (negative) $4GAG_{av}$ value; (Fig. 2). Theoretically,
aggressor (victim) basins have higher (lower) denudation rates than the underlying uplift rate. Therefore graphs in figure 6
must be divided into four quadrants, with aggressors situated in the lower left one, and victims in the higher right one. This
result is verified for ca. 91, 8881 %, 52 % and 8281 % of basins for aggressivity metric based on headwater $\chi$ , slope and
elevation values, $\Delta \chi_{av} \perp \Delta G_{av}$ and $AH_{av}$ , respectively (Figs. 6a, b and e). Several basins depart significantly from the expected
quadrants for $\Delta G$ and $\Delta H_{av}$ , respectively (Figs. 6a, b and e). Several basins depart significantly from the expected
quadrants for $\Delta G$ and $\Delta H_{av}$ , respectively (Higs. 6a, b and e). Several basins depart significant by a linear
relationship (Figs. 6a and b). Part of the dispersion observed around this first order trend may be explained by
more simulations in the calculation of departation metrics fig. 7b). Compared to other metrics $\Delta H$                                                                                                                                                                                                                                                                                                                                                              | Unis)                   |
| reference models after a simulation duration of 2.5 Myr. Denudation rates may be affected by knickpoints, which are a
source of transient perturbation at the scale of the catchment. Therefore, in order to focus only on perturbations associated
with drainage divide dynamics, basins that contain knickpoints are ignored. In agreement with cross-divide metrics tested by
Forte and Whipple (2018), aggressorgraphs in figure 7 must be divided into four quadrants. Aggressor (victim) basins have
negative (positive) $4\chi 4\chi_{av}$ and $4H 4H_{av}$ values and conversely positive (negative) $4G 4G_{av}$ value; (Fig. 2). Theoretically,
aggressor (victim) basins have higher (lower) denudation rates than the underlying uplift rate. Therefore graphs in figure 6
must be divided into four quadrants, with aggressors situated in the lower left one, and victims in the higher right one. This
result is verified for ca. 91, 8881 %, 52 % and 8281 % of basins for aggressivity metric based on headwater $\chi$ , slope and
elevation values, $4\chi_{av} \pm 4G_{av}$ and $4H_{av*}$ respectively (Figs. 6a, b and c). Several basins depart significantly from the expected
quadrants for $4G$ and $4H$ : these eshibit significant knickpoints in their drainage network that increase measured demudation
meters. For this limited dataset, the evolution between $E/I$ and both $4\chi 4\chi_{av}$ and $4G$ cand $4H_{av}$ may be defined by a linear
relationship (Figs. 6a and b). Part of the dispersion observed around this first order trend may be explained by
moreorimptions in the calculation rates and aggressivity metrics Fig. 7b. Compared to other metrics $4H$                                                                                                                                                                                                                                                                                                                                                                                                                                                                                                                                                 |                         |
| source of transient perturbation at the scale of the catchment. Therefore, in order to focus only on perturbations associated
with drainage divide dynamics, basins that contain knickpoints are ignored. In agreement with cross-divide metrics tested by
Forte and Whipple (2018), aggressorgraphs in figure 7 must be divided into four quadrants. Aggressor (victim) basins have
negative (positive) $4\chi d\chi_{av}$ and $4H \Delta H_{av}$ values and conversely positive (negative) $4G\Delta G_{av}$ value-: (Fig. 2). Theoretically,
aggressor (victim) basins have higher (lower) denudation rates than the underlying uplift rate. Therefore graphs in figure 6
must be divided into four quadrants, with aggressors situated in the lower-left one, and victims in the higher-right one. This
result is verified for ca. 91, 8881 %, 52 % and 8281 % of basins for aggressivity metric based on headwater $\chi$ , slope and
elevation values, $d\chi_{av} \perp \Delta G_{av}$ and $\Delta H_{av}$ respectively (Figs. 6a, b and c). Several basins depart significantly from the expected
quadrants for $\Delta G$ and $\Delta H$ : these exhibit significant knickpoints in their drainage network that increase measured denudation
relationship (Figs. 6a and b). Part of the dispersion observed around this first order trend may be defined by a linear
relationship (Figs. 6a and b). Part of the dispersion observed around this first order trend may be explained by
more simplified in the calculation of depudation metrics and acceptersivity metrics. Fig. 7b) Commared to other metrics $4H$                                                                                                                                                                                                                                                                                                                                                                                                                                                                                                                                                                                                                                      |                         |
| with drainage divide dynamics, basins that contain knickpoints are ignored. In agreement with cross-divide metrics tested by
Forte and Whipple (2018), aggressorgraphs in figure 7 must be divided into four quadrants. Aggressor (victim) basins have
negative (positive) $4\chi \Delta \chi_{av}$ and $4H\Delta H_{av}$ values and conversely positive (negative) $4G\Delta G_{av}$ value - (Fig. 2) . Theoretically,
aggressor (victim) basins have higher (lower) denudation rates than the underlying uplift rate. Therefore graphs in figure 6
must be divided into four quadrants, with aggressors situated in the lower left one, and victims in the higher right one. This
result is verified for ca. 91, 8881 %, 52 % and 8281 % of basins for aggressivity metric based on headwater $\chi$ , slope and
elevation values, $\Delta \chi_{av} \Delta G_{av}$ and $\Delta H_{av}$ , respectively (Figs. 6a, b and c). Several basins depart significantly from the expected
quadrants for $\Delta G$ and $\Delta H_{av}$ , respectively (Figs. 6a, b and c). Several basins depart significantly from the expected
quadrants for $\Delta G$ and $\Delta H_{av}$ are spectively (Figs. 6a, b and c). Several basins depart significantly from the expected
quadrants for $\Delta G$ and $\Delta H_{av}$ these exhibit significant knickpoints in their drainage network that increase measured denudation
relationship (Figs. 6a and b). Part of the dispersion observed around this first order trend may be explained by
more simulations in the calculation of depudation rates and aggressivity metrice Fig. 7b. Compared to other metrics $\Delta H_{av}$                                                                                                                                                                                                                                                                                                                                                                                                                                                                                                                                                                                     |                         |
| Forte and Whipple (2018), aggressorgraphs in figure 7 must be divided into four quadrants. Aggressor (victim) basins have
negative (positive) $4\chi \Delta \chi_{av}$ and $4H\Delta H_{av}$ values and conversely positive (negative) $4G\Delta G_{av}$ value: (Fig. 2). Theoretically,
aggressor (victim) basins have higher (lower) denudation rates than the underlying uplift rate. Therefore graphs in figure 6
must be divided into four quadrants, with aggressors situated in the lower left one, and victims in the higher right one. This
result is verified for ca. 91, 8881 %, 52 % and 8281 % of basins for aggressivity metric based on headwater $\chi$ , slope and
elevation values, $\Delta \chi_{av} \perp \Delta G_{av}$ and $\Delta H_{av}$ respectively (Figs. 6a, b and c). Several basins depart significantly from the expected
quadrants for $\Delta G$ -and $\Delta H$ : these exhibit significant knickpoints in their drainage network that increase measured denudation
relationship (Figs. 6a and b). Part of the dispersion observed around this first order trend may be explained by
approximations in the calculation of depudation rates and aggressivity metrics Fig. 7b). Compared to other metrics $\Delta H$                                                                                                                                                                                                                                                                                                                                                                                                                                                                                                                                                                                                                                                                                                                                                                                                                                                                                                                                                                                                                      |                         |
| negative (positive) $4\chi\Delta\chi_{av}$ and $4H\Delta H_{av}$ values and conversely positive (negative) $4G\Delta G_{av}$ value. (Fig. 2). Theoretically,
aggressor (victim) basins have higher (lower) denudation rates than the underlying uplift rate. Therefore graphs in figure 6
must be divided into four quadrants, with aggressors situated in the lower left one, and victims in the higher right one. This
result is verified for ca. 91, 8881 %, 52 % and 8281 % of basins for aggressivity metric based on headwater $\chi$ , slope and
elevation values, $\Delta\chi_{av}$ , $\Delta G_{av}$ and $\Delta H_{av}$ , respectively (Figs. 6a, b and c). Several basins depart significantly from the expected
quadrants for $\Delta G$ and $\Delta H_{iv}$ these exhibit significant knickpoints in their drainage network that increase measured denudation
rates. For this limited dataset, the evolution between $E/U$ and both $4\chi\Delta\chi_{av}$ and $\Delta G$ can $\Delta H_{av}$ may be defined by a linear
relationship (Figs. 6a and b). Part of the dispersion observed around this first order trend may be explained by
approximations in the calculation of depudation mass and aggressivity metrics. Fig. 7b). Compared to other metrics $\Delta H$                                                                                                                                                                                                                                                                                                                                                                                                                                                                                                                                                                                                                                                                                                                                                                                                                                                                                                                                                                                      |                         |
| 285 aggressor (victim) basins have higher (lower) denudation rates than the underlying uplift rate. Therefore graphs in figure 6
must be divided into four quadrants, with aggressors situated in the lower-left one, and victims in the higher-right one. This
result is verified for ca. 91, 8881 %, 52 % and 8281 % of basins for aggressivity metric based on headwater $\chi$ , slope and
elevation values, $\Delta \chi_{av} \Delta G_{av}$ and $\Delta H_{av}$ respectively (Figs. 6a, b and c). Several basins depart significantly from the expected
quadrants for $\Delta G_{and} \Delta H_{av}$ respectively (Figs. 6a, b and c). Several basins depart significantly from the expected
quadrants for $\Delta G_{and} \Delta H_{it}$ these exhibit significant knickpoints in their drainage network that increase measured denudation
rates. For this limited dataset, the evolution between $E/U$ and both $4\chi\Delta\chi_{av}$ and $\Delta G$ can $\Delta H_{av}$ may be defined by a linear
relationship (Figs. 6a and b). Part of the dispersion observed around this first order trend may be explained by
more simulations in the calculation of denudation meas and aggressivity metrics. Fig. 7b). Compared to other metrics $\Delta H$                                                                                                                                                                                                                                                                                                                                                                                                                                                                                                                                                                                                                                                                                                                                                                                                                                                                                                                                                                                                             |                         |
| must be divided into four quadrants, with aggressors situated in the lower-left one, and victims in the higher-right one. This
result is verified for ca. 94, 8881 %, 52 % and 8281 % of basins for aggressivity metric based on headwater $\chi$ , slope and
elevation values, $\Delta \chi_{av}$ , $\Delta G_{av}$ and $\Delta H_{av}$ , respectively (Figs. 6a, b and c). Several basins depart significantly from the expected
quadrants for $\Delta G_{au}$ and $\Delta H_{av}$ , respectively (Figs. 6a, b and c). Several basins depart significantly from the expected
quadrants for $\Delta G_{au}$ and $\Delta H_{av}$ , respectively (Figs. 6a, b and c). Several basins depart significantly from the expected
relationship (Figs. 6a and b). Part of the dispersion observed around this first order trend may be explained by
more vinations in the calculation of depuddion metes and aggressivity metrics. Fig. 7b). Compared to other metrics $\Delta H$                                                                                                                                                                                                                                                                                                                                                                                                                                                                                                                                                                                                                                                                                                                                                                                                                                                                                                                                                                                                                                                                                                                                                                                                                                                                                                       |                         |
| result is verified for ca. 94, 8881 %, 52 % and 8281 % of basins for aggressivity metric based on headwater $\chi$ , slope and
elevation values, $\Delta \chi_{av} \Delta G_{av}$ and $\Delta H_{av}$ respectively (Figs. 6a, b and c). Several basins depart significantly from the expected
quadrants for $\Delta G_{av} \Delta G_{av}$ and $\Delta H_{av}$ respectively (Figs. 6a, b and c). Several basins depart significantly from the expected
quadrants for $\Delta G_{av} \Delta G_{av}$ and $\Delta H_{av}$ respectively (Figs. 6a, b and c). Several basins depart significantly from the expected
quadrants for $\Delta G_{av} \Delta G_{av}$ and $\Delta H_{av}$ respectively (Figs. 6a, b and c). Several basins depart significantly from the expected
relationship (Figs. 6a and b). Part of the dispersion observed around this first order trend may be explained by
approximations in the calculation of depuddion mess and aggressivity metrics. Fig. 7b). Compared to other metrics $\Delta H$                                                                                                                                                                                                                                                                                                                                                                                                                                                                                                                                                                                                                                                                                                                                                                                                                                                                                                                                                                                                                                                                                                                                                                                                                                                             |                         |
| elevation values, $\Delta \chi_{av} \Delta G_{av}$ and $\Delta H_{av}$ respectively (Figs. 6a, b and c). Several basins depart significantly from the expected
quadrants for $\Delta G_{av}$ and $\Delta H_{av}$ respectively (Figs. 6a, b and c). Several basins depart significantly from the expected
quadrants for $\Delta G_{av}$ and $\Delta H_{av}$ respectively (Figs. 6a, b and c). Several basins depart significantly from the expected
rates. For this limited dataset, the evolution between $F/U$ and both $\Delta \chi \Delta \chi_{av}$ and $\Delta G$ can $\Delta H_{av}$ may be defined by a linear
relationship (Figs. 6a and b). Part of the dispersion observed around this first order trend may be explained by
approximations in the calculation of depuddion rates and aggressivity metrics. Fig. 7b). Compared to other metrics $\Delta H$                                                                                                                                                                                                                                                                                                                                                                                                                                                                                                                                                                                                                                                                                                                                                                                                                                                                                                                                                                                                                                                                                                                                                                                                                                                                                                                                                                                                               |                         |
| quadrants for $\Delta G$ and $\Delta H$ : these exhibit significant knickpoints in their drainage network that increase measured denudation
rates. For this limited dataset, the evolution between $E/U$ and both $\Delta_X \Delta_{Xav}$ and $\Delta G$ can $\Delta H_{av}$ may be defined by a linear
relationship (Figs. 6a and b). Part of the dispersion observed around this first order trend may be explained by
approximations in the calculation of depudation mass and accessivity metrics Fig. 7b). Compared to other metrics $\Delta H$                                                                                                                                                                                                                                                                                                                                                                                                                                                                                                                                                                                                                                                                                                                                                                                                                                                                                                                                                                                                                                                                                                                                                                                                                                                                                                                                                                                                                                                                                                                                                                                                                                                                                                                                     |                         |
| 290 rates. For this limited dataset, the evolution between $E/U$ and both $4\chi \Delta \chi_{ap}$ and $4G$ can $\Delta H_{ap}$ may be defined by a linear relationship (Figs. 6a and b). Part of the dispersion observed around this first order trend may be explained by Mis enforme: Anglais(États) more simulations in the calculation of depuddion rates and accreasivity metrics Fig. 7b). Compared to other metrics $4H$                                                                                                                                                                                                                                                                                                                                                                                                                                                                                                                                                                                                                                                                                                                                                                                                                                                                                                                                                                                                                                                                                                                                                                                                                                                                                                                                                                                                                                                                                                                                                                                                                                                                                                                                                                                                                                                                  |                         |
| relationship (Figs. 6a and b). Part of the dispersion observed around this first order trend may be explained by Mis enforme: Anglais(États